# TRANSLATING FLOW TO POLICY VIA HINDSIGHT ONLINE IMITATION

**Yitian Zheng**[1,*]**, Zhangchen Ye**[1,*]**, Weijun Dong**[2,*]**, Shengjie Wang**[1]**, Yuyang Liu**[1]**,
Chongjie Zhang**[3]**, Chuan Wen**[4,†]**, Yang Gao**[1,5,†]

[1]Institute for Interdisciplinary Information Sciences, Tsinghua University
[2]University of California San Diego [3]Washington University in St. Louis
[4]Shanghai Jiao Tong University [5]Shanghai Qi Zhi Institute
{zhengyt23,yezc22}@mails.tsinghua.edu.cn, weijundong@ucsd.edu
{wangsj23,yyliu22}@mails.tsinghua.edu.cn, chongjie@wustl.edu
wenchuan@sjtu.edu.cn, gaoyangiiis@mail.tsinghua.edu.cn

## ABSTRACT

Recent advances in hierarchical robot systems leverage a high-level planner to propose task plans and a low-level policy to generate robot actions. This design allows training the planner on action-free or even non-robot data sources (e.g., videos), providing transferable high-level guidance. Nevertheless, grounding these high-level plans into executable actions remains challenging, especially with the limited availability of high-quality robot data. To this end, we propose to improve the low-level policy through online interactions. Specifically, our approach collects online rollouts, retrospectively annotates the corresponding high-level goals from achieved outcomes, and aggregates these hindsight-relabeled experiences to update a goal-conditioned imitation policy. Our method, Hindsight Flow-conditioned Online Imitation (HinFlow), instantiates this idea with 2D point flows as the high-level planner. Across diverse manipulation tasks in both simulation and physical world, our method achieves more than $2\times$ performance improvement over the base policy, significantly outperforming the existing methods. Moreover, our framework enables policy acquisition from planners trained on cross-embodiment video data, demonstrating its potential for scalable and transferable robot learning.

**Project page:** https://dwjshift.github.io/HinFlow

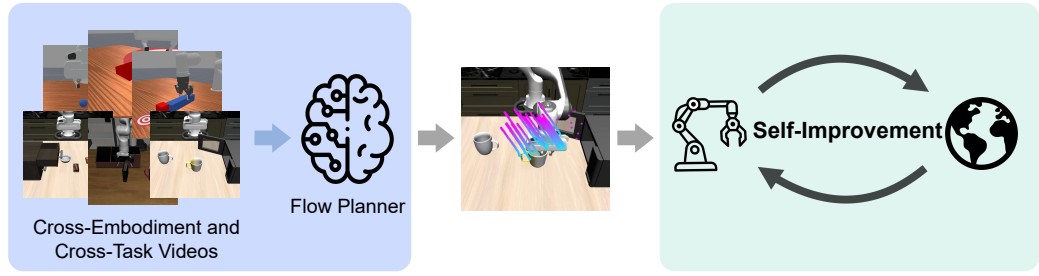

Figure 1: **Motivation of HinFlow.** Learning from large-scale and diverse video data, a flow-based high-level planner can formulate generalizable task plans. To robustly execute these plans, the low-level control policy needs to be iteratively refined through online practice, preventing the overall system from being bottlenecked by low-level execution.

---

*Equal contribution.
†Equal advising. Corresponding authors.

# 1 INTRODUCTION

The success of vision and language models is deeply rooted in their ability to train on vast quantities of data (Brown et al., 2020; Ravi et al., 2024). However, in robotics, this reliance on large datasets presents a key challenge for traditional end-to-end approaches like imitation learning (Lin et al., 2024). Consequently, hierarchical robotic systems have garnered considerable interest as a way to circumvent this fundamental data bottleneck. These systems structure robotic control problems into two hierarchical levels, where a high-level planner decomposes abstract task specifications into subgoals to guide the low-level controller. This decomposition separates the high-level planner from the robot's complex physical dynamics, thereby enabling it to be trained on vast, action-free datasets, even including non-robot data.

Flow of points is a general representation of these high-level plans, describing the predicted state as future keypoint trajectories (Wen et al., 2023; Xu et al., 2025; Haldar & Pinto, 2025). Since point flow explicitly encodes physical motion dynamics and is robust to variations in visual appearance, it offers compact and effective guidance for the low-level controller. However, translating these flow plans into robust and scalable low-level policies remains a critical challenge. A direct approach is using analytical or optimization-based methods to compute actions, but they often struggle with real-world complexities, e.g., visual occlusions or non-rigid-body dynamics (Bharadhwaj et al., 2024b; Yuan et al., 2024). Alternatively, learning a data-driven low-level policy is a more general method; however, it typically relies on collecting high-quality, in-domain robot data (Wen et al., 2023; Gao et al., 2024), which is an expensive and laborious process that undermines the goal of scalable skill acquisition.

To address these challenges, we propose Hindsight Flow-conditioned Online Imitation (HinFlow), a simple yet effective approach that enables the robot to ground the high-level flows into executable policies through self-practice. The robot interacts with the environment under the flow guidance to generate exploratory behaviors and continuously refine a flow-conditioned control policy. The core insight is that even if the collected experiences fail to reach the planner's precise goal, they can be repurposed for self-imitation by framing the achieved flows as the intended goal (Ghosh et al., 2019). Generating such supervision directly from the robot's own imperfect experiences allows it to learn and adapt without relying on a large volume of expert data, thus constructing a scalable bridge from flow plans to robust physical execution.

We demonstrate the effectiveness of HinFlow through experiments on benchmark tasks from LIBERO (Liu et al., 2023) and ManiSkill (Tao et al., 2025). Our key contributions are summarized as follows:

1. We propose the Hindsight Flow-conditioned Online Imitation (HinFlow) framework, which grounds the high-level plans learned from action-free videos into a robust low-level policy through online imitation.

2. HinFlow achieves an average success rate of $84.0\%$ in seven manipulation tasks, outperforming the strongest baseline by a factor of $1.45\times$. Crucially, this result is achieved with only 80K online interaction steps, showcasing the high sample efficiency of our approach. (Section 5.2) We also validate HinFlow's efficiency and reliability in a real-world experiment. (Section 5.3)

3. Moreover, HinFlow demonstrates remarkable versatility and robustness. It can effectively improve policies by learning from cross-embodiment videos. The learned flow-conditioned policy is robust to visual variations and achieves zero-shot generalization to novel objects and distractors. (Section 5.4)

# 2 RELATED WORK

## 2.1 HIERARCHICAL ROBOTIC SYSTEMS

Hierarchical robotic systems structure control problems into a high-level planner, which generates subgoal guidance, and a low-level controller tasked with achieving them. Previous works have explored various goal representations, such as future images (Du et al., 2023; Black et al., 2023; Yang et al., 2023), point flows (Wen et al., 2023; Bharadhwaj et al., 2024a), motion fields (Yin et al., 2025), or learned neural representations (Wang et al., 2023; Fang et al., 2023). Despite this

diversity in goal formats, methods for mapping them to low-level control mainly fall into a few categories. One approach uses analytic or optimization-based controllers that, for example, compute an optimal rigid-body transform to align the current state with the goal (Ko et al., 2023; Yuan et al., 2024; Yin et al., 2025). While direct, such methods inherently rely on assumptions like object rigidity. Another thread is a data-driven approach, which learns policy models for low-level control. Although general, prior work typically trains a goal-conditioned policy or inverse dynamics model on high-quality, in-domain robot data (Wang et al., 2023; Du et al., 2023; Wen et al., 2023), a process that requires substantial human effort in data collection and limits scalability. Consistent with our approach, recent work also considers involving online interactions to improve policy. For example, Escontrela et al. (2023) uses high-level video prediction to construct rewards and trains policies via reinforcement learning (RL). However, RL with visual rewards typically suffers from inefficient exploration and the complexities of optimizing long-horizon rewards (Liu et al., 2024). In contrast, we frame policy learning as supervised learning on hindsight-relabeled examples. This yields a simple, well-conditioned objective that is easy to optimize. Moreover, Luo & Du and Zhou et al. (2025) both utilize hindsight relabeling to enhance goal-conditioned policies: Luo & Du employs video diffusion models to guide online exploration, while Zhou et al. (2025) leverages vision-language models to generate task goal proposals. Unlike their image-goal formulation, we leverage point flows as a high-level plan, which is a compact, low-dimensional encoding of task-relevant motion. This abstraction filters out non-task-relevant visual variations, leading to a more robust control policy.

## 2.2 FLOW-BASED MANIPULATION

Point-motion (flow) extraction from visual inputs offers a general feature representation. Leveraging this representation, prior work has successfully guided manipulation tasks using point flows derived from diverse sources. Some approaches train flow prediction models on action-free or in-the-wild video datasets (Wen et al., 2023; Yuan et al., 2024; Bharadhwaj et al., 2024b). An alternative stream facilitates learning from human demonstrations by adopting object-centric flows (Xu et al., 2025; Wang et al., 2025) or unifying keypoint representations between human hands and robot grippers (Haldar & Pinto, 2025; Ren et al., 2025). Gao et al. (2024) treats flow as an action representation within a world model, performing model-based planning to derive the optimal flow. Gu et al. (2023) further explores using manually drawn or LLM-generated trajectory sketches to provide effective guidance.

Once the flows are obtained, the subsequent flow-to-policy translation mirrors the paradigms outlined in Section 2.1. Our approach is closely related to research on flow-based reinforcement learning. Guzey et al. (2025) leverages object point flows in a human video, defining a reward that encourages the agent to replicate these motion trajectories. Similarly, Yu et al. (2025) uses a flow derived from a generative model to shape an object-centric dense reward function. However, unlike prior methods that require predicting the full, long-horizon flow (point motions across an entire rollout), we utilize short-horizon flows as the high-level goal. While this difference might appear small, it has two major implications: (1) It is a significant challenge for the high-level model to predict the long-horizon flow. (2) Furthermore, using short-horizon flows allows us to reframe the problem as self-imitation. In our framework, the policy learns to replicate the achieved flow sequences from collected experiences, offering an intuitive and effective policy improvement algorithm.

## 3 PRELIMINARY

In this paper, we consider the challenge of learning a hierarchical policy from two complementary sources: a large-scale action-free video dataset and a small-scale set of action-labeled expert demonstrations. The videos provide high-level task knowledge, which we leverage to train a planner that generates subgoals, while the demonstrations are used to learn a low-level policy that maps the guidance to concrete actions. We introduce the notations below.

**Policy Learning from Videos and Demonstrations.** The action-free video dataset is denoted by $\mathcal{D}_h = \{\tau_o^i\}_{i=1}^{N_h}$, where $N_h$ denotes the number of videos. Following existing research (Wen et al., 2023), $\mathcal{D}_h$ is leveraged to train a high-level planner that produces subgoals $\mathcal{G}_t$ at each timestep $t$. In addition, we assume access to an action-labeled dataset $\mathcal{D}_a = \{\tau_a^i\}_{i=1}^{N_a}$, where each trajectory $\tau_a^i = \{(o_t^i, a_t^i)\}_{t=1}^T$ consists of observation–action pairs of length $T$. These demonstrations are

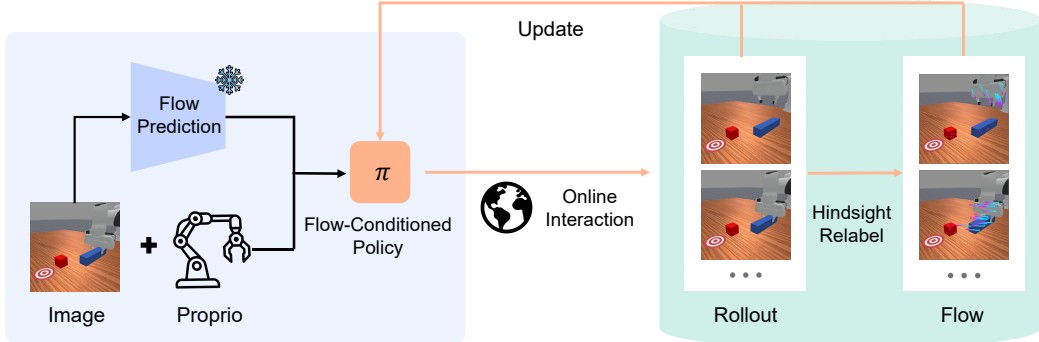

Figure 2: **Overview of HinFlow.** Left: Hierarchical Policy. Our framework employs a flow prediction model to generate high-level plans in the form of point flow, which guide the low-level policy. Right: Hindsight Relabeled Replay Buffer. The robot rollouts the policy in the environment to collect explorative trajectories and retrospectively annotate the achieved flow subgoals using a video tracker. Subsequently, it performs policy updates conditioned on hindsight-relabeled flows, which in turn creates a virtuous cycle for Self-improvement.

typically employed to train a low-level imitation policy $\pi$ to execute the task under the guidance of the high-level planner. Formally, the policy parameters $\theta$ are optimized as follows:

$$\min_{\theta} \ \mathbb{E}_{(o_t,a_t)\sim\mathcal{D}_a}\big[\mathcal{L}(\pi(o_t,\mathcal{G}_t;\theta),a_t)\big], \tag{1}$$

where $\mathcal{L}$ denotes the loss function for imitation learning, e.g, mean squared error (MSE).

**Point Flow.** We adopt *point flow* as our high-level plan representation. Point flow captures motion patterns from videos while being invariant to task-irrelevant factors like appearance and lighting. To extract point flow from videos, we first select a set of initial points $\mathbf{p}_0 = \{p_{0,k}\}_{k=1}^{K}$ in the first camera frame, where $p_{0,k} = (x_{0,k}, y_{0,k})$ denotes pixel coordinates. We then apply an off-the-shelf video tracker (Karaev et al., 2024b) to obtain their 2D trajectories $\mathbf{p}_t$ for $t = 1, \ldots, T$. These trajectories constitute the ground-truth labels for training the high-level planner. The trained planner predicts the subgoal at time $t$ in the format of $\mathcal{G}_t = \{\hat{\mathbf{p}}_i\}_{i=t}^{t+H}$, where $H$ denotes the planning horizon.

## 4 METHODOLOGY

Video data can enhance policy learning by providing a high-level planner with prior knowledge of task-relevant behaviors and motion patterns. Point flow is a natural representation for such high-level plans, as it captures motion dynamics from videos while filtering out control-irrelevant factors. Nevertheless, learning a low-level policy that reliably maps flow-based guidance to robot actions remains challenging, particularly when demonstrations are limited. Motivated by this, we propose *Hindsight Flow-conditioned Online Imitation* (HinFlow), a framework that leverages online interaction to improve low-level imitation learning.

### 4.1 HIGH-LEVEL PLANNER WITH FLOW MODELING FROM VIDEOS

We adopt point flow as the representation for the high-level planner, leveraging its capability to provide dense subgoals $\mathcal{G}_t$ at each timestep $t$. Specifically, the flow prediction model forecasts the future trajectories of task-relevant points based on the current images, thereby generating a sequence of subgoals. This process can be formally expressed as: $\mathcal{G}_t = \mathbf{F}_{\text{flow}}(o_t, \mathbf{p}_t)$ where $\mathbf{F}_{\text{flow}}$ parameterized by $\xi$ denotes the flow prediction function and $\mathbf{p}_t$ indicates the current positions of the query points.[1] By predicting the positions of points for the next $H$ frames, the model ensures that the high-level planner provides comprehensive guidance for task execution.

---

[1]While our implemented flow prediction model is language-conditioned to incorporate data from various tasks, we omit this detail in the main text for simplicity.

---

**Algorithm 1** Hindsight Flow-conditioned Online Imitation (HinFlow)

---

**Require:** Action-free video dataset $\mathcal{D}_h$, Action-labeled dataset $\mathcal{D}_a$, Video tracker $\Phi$
 1: Train flow prediction model $\mathbf{F}_{\text{flow}}$ with $\mathcal{D}_h$                                             ▷ Equation (2)
 2: Pretrain flow-conditioned policy $\pi$ with $\mathcal{D}_a$
 3: Initialize replay buffer $\mathcal{D}_r$
 4: **for** episode $= 1, 2, \cdots$ **do**
 5:      Sample a trajectory $\tau = \{o_1, a_1, \cdots, o_T\} \sim \mathbf{F}_{\text{flow}} \circ \pi$
 6:      Use $\Phi$ to compute achieved flows in $\tau$
 7:      Add flow-annotated tuples $(o_t, a_t, \{\mathbf{p}_i\}_{i=t}^{t+H})$ to $\mathcal{D}_a$
 8:      Sample batch from $\mathcal{D}_r$ and update $\pi_\theta$                                     ▷ Equation (3)
 9: **end for**

---

To enhance the guidance with rich manipulation-relevant information, we strategically select the query points $\mathbf{p}_t$ using a task-centric point sampler. Specifically, for the third-person camera, we segment the end effector and key objects to identify regions pertinent to task execution, and then randomly sample points within these regions. This approach ensures that the flow prediction model is trained on task-relevant inputs, thereby enhancing its accuracy and generalizability. For the wrist-mounted camera, since the task-relevant objects may not consistently appear in this view, we simply use a fixed set of 32 points on a grid.

To realize the flow-based planner, we first extract flow information from the video dataset $\mathbf{D}_h$ for the high-level training process. As detailed in Section 3, we use an off-the-shelf video tracker $\Phi$ (Karaev et al., 2024a) to generate flow labels for the video dataset. Specifically, for each observation $o_t$ in $\mathcal{D}_h$, we apply our task-centric point sampler to generate a set of query points $\mathbf{p}_t$ and then track their future trajectory $\mathbf{p}_{t+1}^{t+H}$. Such data tuple $(o_t, \mathbf{p}_t^{t+H})$ are stored into the annotated video dataset $\bar{\mathcal{D}}_h$. Further data augmentation details are provided in Appendix A.1.

After obtaining the flow annotations, we proceed to train the flow prediction model. Following the Track Transformer model in ATM (Wen et al., 2023), we tokenize the images and query points, and feed them into a multi-modal transformer model. The model is trained using a flow prediction loss, which can be formally expressed as:

$$\min_{\xi} \; \mathbb{E}_{(o_t, \mathbf{p}_t^{t+H}) \sim \bar{\mathcal{D}}_h} \left[ \mathcal{L}_{\text{flow}} \left( \mathbf{F}_{\text{flow}}(o_t, \mathbf{p}_t; \xi), \mathbf{p}_{t+1}^{t+H} \right) \right] \tag{2}$$

where $\mathcal{L}_{\text{flow}}$ denotes the flow prediction loss function, and $\bar{\mathcal{D}}_h$ represents the annotated video dataset. This training objective ensures that the model accurately predicts the future trajectories of task-relevant points, thereby enhancing the high-level planner's ability to generate effective subgoals.

### 4.2 Low-level Policy with Hindsight Online Imitation

Based on the high-level planner, our objective is to ground the implicit motion information in the flow into an executable low-level policy. This low-level policy can be formulated as a flow-conditioned policy $\pi(o_t, \mathbf{F}_{\text{flow}}(o_t, \mathbf{p}_t))$. Although the high-level planner can provide robust and generalizable flows for various scenarios, the policy's capacity to achieve each subgoal is limited by the scarcity of action-labeled data.

We propose an approach to overcome this limitation. As shown in Figure 2, the robot interacts with the environment under the guidance of the flow planner to generate exploratory behaviors, which in turn continuously refine its low-level policy $\pi$. To obtain the necessary supervision signals, we extract the actual point flows from the robot's own rollout videos. These achieved flows then serve as the goal labels for training $\pi$. This self-supervised loop, which generates training data directly from the robot's imperfect experiences, enables the system to learn and adapt without relying on large volumes of expert demonstrations.

**Online data collection and labeling.** To collect data for policy training, the robot interacts with the environment using its hierarchical policy. To facilitate exploration and enable the policy to encounter novel states, we introduce a modest amount of exploration noise into the action, as detailed in Appendix A.1. The environment responds to the output action by providing a new observation and state. This process is iterated until the episode terminates. After recording one full episode

$\{(o_t, a_t)\}_{t=1}^{T}$, the pipeline conducts hindsight relabeling (Kaelbling, 1993; Andrychowicz et al., 2017; Ghosh et al., 2019; Yang et al., 2022) to produce valid training data. To annotate these data with flow labels, we utilize the video tracker $\Phi$ to process the collected data, generating the achieved flow. The point sampling strategy and data augmentation are consistent with those described in Section 4.1. The labeled data $(o_t, a_t, \{\mathbf{p}_i\}_{i=t}^{t+H})$ for $t = 1, \ldots, T$ are subsequently incorporated into the replay buffer $\mathcal{D}_r$.

**Policy model training.** The architecture of our flow-conditioned low-level policy adheres to the transformer-based design established in prior research (Kim et al., 2021; Wen et al., 2023). Specifically, the inputs of the current timestep, comprising both visual and proprioceptive data, are initially encoded into a transformer. These input modalities are subsequently encoded into spatial tokens through a spatial transformer. We then combined flow tokens with the spatial tokens to effectively integrate the guiding information from the tracks. The combined tokens are then passed through a series of MLPs to generate actions. Furthermore, we incorporate action chunking into the policy learning process, as elaborated in Appendix A.1.

During the online interaction process, the policy is iteratively optimized using data from the replay buffer $\mathcal{D}_r$. This simultaneous interaction and optimization allow the model to continuously update its behavior based on feedback from the environment. For training, the flow is directly sourced from the dataset, rather than that provided by the high-level planner during inference. The optimization process is formalized as:

$$\min_{\theta} \mathbb{E}_{(o_t, a_t, \{\mathbf{p}_i\}_{i=t}^{t+H}) \sim \mathcal{D}_r} \left[ \mathcal{L} \left( \pi(o_t, \{\mathbf{p}_i\}_{i=t}^{t+H}; \theta), a_t \right) \right] \tag{3}$$

We initialize the low-level policy using the available action-labeled expert demonstrations. The training objective aligns with Equation (3), using data sampled from the offline demonstrations $\mathcal{D}_a$. This pretrained policy serves as a capable starting point, facilitating the collection of meaningful rollouts during the initial exploration phase.

In summary, as presented in Algorithm 1, this framework effectively incorporates flow into actionable policies by iteratively refining the policy through online interactions and hindsight relabeling, enhancing policy stability and generalization.

## 5 EXPERIMENTS

We evaluate the performance and efficiency of HinFlow through comprehensive experiments on seven manipulation tasks. Following a detailed description of the experimental setup in Section 5.1, we present our main results in Section 5.2, comparing HinFlow with four baselines. In Section 5.3, we test the method's efficiency and reliability in the physical world. Furthermore, in Section 5.4, we demonstrate HinFlow's robustness by assessing its capacity for cross-embodiment learning and generalization to unseen visual variations. Finally, a series of ablation studies in Section 5.5 investigates the influence of pivotal parameters. To ensure the reliability of our results, all reported data in this section are averaged over 5 random seeds.

### 5.1 EXPERIMENT SETUP

**Tasks.** To rigorously assess HinFlow's capabilities, we conduct experiments on seven tasks from two benchmarks. The suite includes four tasks from LIBERO (Liu et al., 2023): *Place Butter*, *Place Book*, *Hide Chocolate*, *Close Microwave*, and three from ManiSkill3 (Tao et al., 2025): *Place Sphere*, *Pull Cube Tool*, and *Poke Cube*. This collection presents a diverse set of challenges that span long-horizon manipulation (e.g., *Hide Chocolate*) and complex object interaction (e.g., *Pull Cube Tool*). Figure 3 illustrates the initial and target state for each task.

For each task, the robot's observations consist of images from an external camera and a wrist-mounted camera (both at 128x128 resolution), along with its proprioceptive states. For training, we assume a limited set of action-labeled demonstrations (one for each LIBERO task, five for each ManiSkill task) and a large collection of unlabeled videos covering related tasks in the same scene. The robot is allowed to collect data through online interaction, but the environment will not provide any reward or success signals.

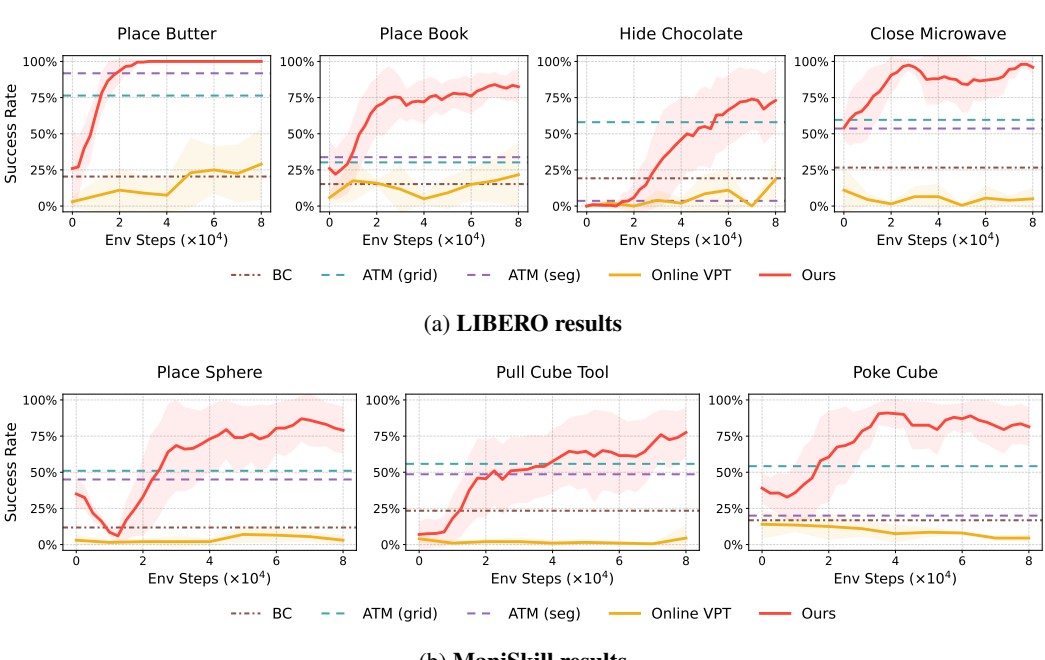

Figure 3: **Visualization of all experiment tasks.** The first four tasks are on LIBERO and the other three are on ManiSkill. The arrows indicate the sequential steps required to complete each task.

(a) **LIBERO results**

(b) **ManiSkill results**

Figure 4: Performance comparison of HinFlow against baselines on the LIBERO and ManiSkill tasks. Online methods interact with the environment for 80000 steps. The shaded region represents the standard deviation across five random seeds. Notably, our method achieves significantly higher performance and sample efficiency.

**Baselines.** We compare HinFlow with the following baselines:

- **BC** denotes a traditional behavior cloning strategy, which trains an observation-to-action policy only on the limited action-labeled demonstrations.

- **ATM** (Wen et al., 2023) first pretrains a track transformer on a large-scale video dataset to predict the future flow of the query points. The official implementation queries a set of points on a fixed grid, which we denote as **ATM (grid)**. We further implement a variant named **ATM (seg)**, which is equipped with our point sampling strategy that concentrates on the robot end-effector and task-relevant objects.

- **Online VPT** adapts the original Video PreTraining (VPT) (Baker et al., 2022) to our setting. VPT first trains an inverse dynamics model (IDM) on action-labeled demonstrations. The trained IDM is then used to provide action labels for a large-scale video dataset, from which a BC policy is learned. Our primary modification is an iterative online refinement loop: it uses online rollouts from the policy to fine-tune the IDM, then uses the improved IDM to relabel the dataset and retrain the policy.

## 5.2 MAIN RESULTS

As shown in Figure 4, HinFlow outperforms all baselines, achieving an average success rate of 84% on all tasks within only 80K environment steps ($300 \sim 400$ episodes). The performance of all

three offline baselines is constrained by the number of action-labeled demonstrations. In contrast, we can observe that online imitation in HinFlow significantly improves the success rate of the base policy. This is particularly evident in challenging tasks like *Hide Chocolate* and *Pull Cube Tool*, where HinFlow boosts the policy's performance from near-zero to an average success rate of 75%. While Online VPT achieves marginal improvements on *Place Book* and *Place Butter*, it struggles significantly on others. Our empirical analysis reveals that its inverse dynamics model introduces substantial errors when pseudo-labeling action-free video data. HinFlow circumvents this issue by using point flow, which provides a more compact and robust goal representation, leading to consistently reliable policy execution.

## 5.3 REAL WORLD EXPERIMENT

In this section, we conduct a real-world experiment to further strengthen our claim.

**Setup.** As shown in Figure 5, the task requires a Franka Emika Panda arm with a Robotiq gripper to grasp and place a mouse onto a pad. The robot operates at a control frequency of 10 Hz, with an action space consisting of the end-effector translations and the gripper command. We employ a wrist-mounted ZED2 camera and a fixed third-person ZED2 camera to capture the images. The RGB images are then cropped and resized to $128 \times 128$ resolution as observations. We leverage Grounded SAM 2 (Ravi et al., 2024; Ren et al., 2024) and CoTracker3 (Karaev et al., 2024a) to extract point flows. For each episode, the mouse is manually reset to a randomized position within a 15cm $\times$ 15cm area (red box in Figure 5).

Data collection is performed via VR teleoperation using a Meta Quest controller, following the hardware and system setup established in DROID (Khazatsky et al., 2024). A dataset of 100 action-free videos is used for training the high-level planner, and only 2 action-labeled expert demonstrations are used to pretrain the low-level policy.

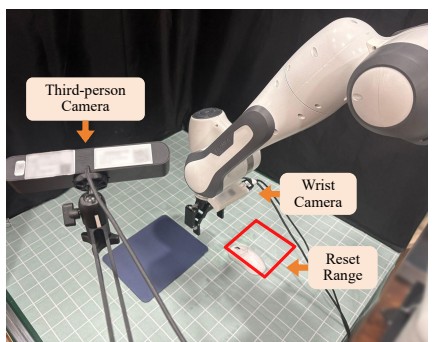

Figure 5: **Visual setup of the real-world experiment.** The robot is required to pick up a mouse and place it on a pad. We highlight the positions of the wrist-mounted and third-person cameras. The red bounding box indicates the 15cm $\times$ 15cm region used for randomized object initialization during resets.

**Results.** We allow the robot to interact with the environment for 10,000 steps (86 episodes, $\sim$1 hour of interleaved data collection and model updates). We compare HinFlow against two baselines: **BC** and **ATM (seg)**. The quantitative results are summarized in Table 1. Restricted by the number of action-labeled demonstrations, both BC and ATM struggle to generalize, succeeding only when the initial state closely resembles the demonstrations. In contrast, while HinFlow starts with an initial success rate of 40%, the online interaction enables the policy to rapidly adapt. With a limited budget of 86 online episodes, HinFlow achieves a final success rate of 95%, demonstrating its efficiency and reliability in the physical world.

| Method | Success |
|--------|---------|
| BC | 4/20 |
| ATM (seg) | 8/20 |
| Ours | 8/20 → 19/20 |

Table 1: **Results of the real world experiment.** We report success rates over 20 evaluation rollouts. The arrow indicates the improvement achieved by HinFlow after 10,000 steps of online interaction.

## 5.4 TRANSFER EXPERIMENTS

In this section, we evaluate HinFlow's robustness along two key dimensions: (1) its ability to learn from cross-embodiment videos, and (2) its low-level policy's generalization capability to unseen distractors or novel target objects.

| Task | Cross-embodiment Data | Success Rate(%) |
|------|-----------------------|-----------------|
| Place Book | w/ | **48.1** |
| | w/o | 0.6 |
| Poke Cube | w/ | **61.3** |
| | w/o | 24.4 |

(a) Franka to Kinova  (b) Franka to xArm  (c) Results

Figure 6: **Cross embodiment transfer.** A high-level planner is trained using a large action-free dataset from a source arm (Franka), combined with only 5 labeled demonstrations from a target arm. Then HinFlow learns a control policy under the planner's guidance. (a) In *Place Book*, the target arm is Kinova. (b) In *Poke Cube*, the target arm is xArm. (c) HinFlow can effectively leverage information from cross-embodiment video data to learn a stronger policy.

**Cross-embodiment transfer.** We consider the following cross-embodiment transfer setting. Our framework is provided with five action-labeled demonstrations from a target robot arm, along with hundreds of action-free videos collected from a different robot arm. We train a high-level planner using these two datasets. Subsequently, HinFlow learns on the target robot arm through online interaction. For comparison, we also test the performance when the high-level planner is trained solely on the five same-embodiment demonstrations. We evaluate this approach on two tasks: *Place Book*, where the target arm is a Kinova Gen3, and *Poke Cube* with an xArm6 as the target arm. For both tasks, a Franka Panda arm serves as the source embodiment. As illustrated in Figure 6, leveraging the cross-embodiment data shows a gain of over 40 percentage points in success rate. These results demonstrate that HinFlow successfully extracts valuable knowledge from action-free cross-embodiment videos, translating it into a significantly more robust policy.

**Policy generalization.** In *Place Butter*, we test two environmental changes: (1) Extra distractors are added to the scene. (2) The butter is replaced with chocolate pudding. A single high-level planner is trained on data from both the original scene and the two modified scenes. The low-level policy is trained on the original task and is directly evaluated on the two modified setups without any training or interaction. For comparison, we train a BC baseline on 10 demonstrations from the original task and evaluate its performance. As shown in Table 2, the BC baseline's performance collapses from 67.5% to under 10%. In contrast, HinFlow's low-level policy proves robust to these environmental perturbations, achieving over a 90% success rate in both experiments. We attribute this robustness to the compact goal representation provided by the flow planner, which abstracts away task-irrelevant visual factors.

| Task | Ours | BC |
|------|------|-----|
| Orginal | 100.0 | 67.5 |
| Extra Distractors | 92.8 | 0.0 |
| Unseen Target Object | 96.2 | 6.5 |

Table 2: **Success rates (%) on policy generalization experiments.** The low-level policy trained on *Place Butter* is zero-shot evaluated under two environment modifications: (1) extra distractors, (2) unseen target object.

### 5.5 ABLATION STUDIES

We conduct ablation studies to analyze the effect of our design choices, including the number of low-level pretraining demonstrations and the flow length. For these experiments, we select two representative tasks from LIBERO and two from ManiSkill, respectively.

**Number of action-labeled demos.** We conduct evaluations using 0, 1, and 3 pretrain demos on LIBERO, and 0, 2, 5, and 10 pretrain demos on ManiSkill. As illustrated in Figure 7, in the absence of any action-labeled data (0 demos), the low-level policy fails to explore meaningful trajectories, resulting in a success rate close to zero. When at least one action-labeled demonstration is available, the initial success rate improves as the number of demos increases. However, after online imitation learning, HinFlow achieves comparable final performance across these conditions. This suggests that HinFlow is not sensitive to the initial policy's quality.

**Flow length.** We investigate the impact of the high-level predicted flow length by evaluating flow lengths of 4, 8, 12, and 16 on both LIBERO and ManiSkill. The results in Figure 8 indicate that

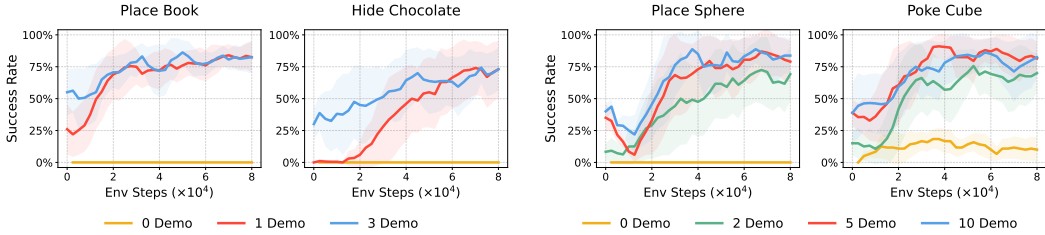

Figure 7: **Ablation study on the number of action-labeled demonstrations.** Left: Results for two LIBERO tasks. Right: Results for two ManiSkill tasks.

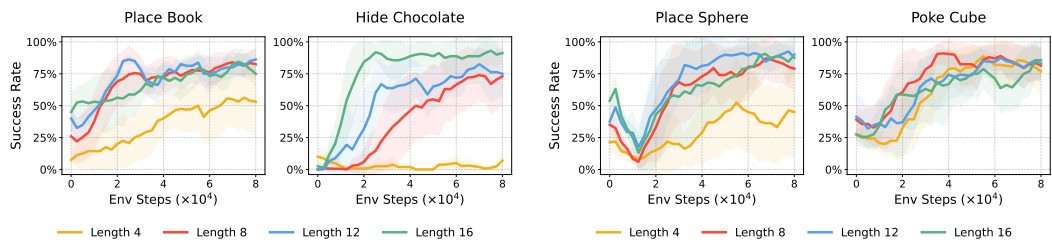

Figure 8: **Ablation study on the flow length.** Flow length is set to 8 by default in our method. Left: result of 2 LIBERO tasks. Right: result of 2 ManiSkill tasks.

flow lengths of 8, 12, and 16 offer stable and robust performance across these tasks, whereas a flow length of 4 results in a significant performance drop. We attribute this drop to the insufficient guidance provided by the overly short horizon.

## 6 CONCLUSION, LIMITATIONS AND FUTURE WORK

In this work, we study the critical challenge of grounding high-level robotic plans, such as point flows, into executable policies. We introduce Hindsight Flow-conditioned Online Imitation (HinFlow), which enables a robot to learn such low-level policies from self-practice. The robot collects exploratory trajectories in the environment, and continuously refines its control policy from such imperfect interactions. The core of our approach is to repurpose the achieved flows as hindsight goals, which provide dense supervision signals for policy learning. Our experiments demonstrate that HinFlow is both sample-efficient and robust.

Despite its effectiveness, our work has several limitations. First, to ensure collecting meaningful rollouts in the early exploration phase, we initialize our low-level policy with a small number of in-domain, action-labeled demonstrations. Recent research on vision-language-action models with open-world capabilities (Zitkovich et al., 2023; Kim et al., 2024; Black et al., 2024) could potentially provide these initial exploratory trajectories, thereby eliminating the need for any action-labeled data. Second, our work is built upon 2D point flow, which is inherently ambiguous for complex 3D motions. Therefore, extending our method to more advanced goal representations, such as 3D motion fields (Yin et al., 2025), presents a valuable direction for future investigation.

## 7 REPRODUCIBILITY STATEMENT

To ensure reliability, all experiments are repeated over 5 independent random seeds, and we report the mean ± standard deviation. Complete implementation details for HinFlow and all baselines are provided in Appendix A to facilitate reproduction.

## ACKNOWLEDGEMENT

This work is supported by Shanghai Qi Zhi Institute & Spirit AI Innovation Program and the Tsinghua University Dushi Program.

We thank Chengbo Yuan for assistance with the robot hardware setup, and Ruiqian Nai for help in managing the computing resources.

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

# A  IMPLEMENTATION DETAILS

## A.1  IMPLEMENTATION DETAILS OF OUR METHOD

**Policy Architecture.**  In our experiments, the low-level policy is designed as a transformer-based architecture. The input to our policy includes a set of temporally stacked images across multiple views $o_t \in \mathbb{R}^{V \times T \times C \times H \times W}$ and proprioceptive data $p_t \in \mathbb{R}^{D_p}$. These inputs are processed through three main stages. Initially, we encode the image data into spatial tokens at each timestep. These tokens are concatenated across all views with a learned spatial CLS token, and self-attention is applied to the sequence. The spatial CLS token is then extracted as the representation of the spatial information. Following the spatial encoding, the proprioceptive data at each timestep is projected into the same shared embedding space $\mathbb{R}^D$. The encoded proprioceptive data, the spatial CLS token, and a learned action CLS token are then interleaved across timesteps into a sequence. Causally-masked self-attention is performed on this sequence to capture temporal dependencies. Finally, each timestep's output is treated independently with an MLP, which parameterizes actions by taking the action CLS token and fusing it with the reconstructed flows of the current timestep.

**Point Sampling.**  In our pipeline, a crucial step in obtaining flow from video is sampling the task-relevant points within the image coordinates. In simulation experiments, we first acquire segmentation masks directly from the simulation environment and subsequently identify the regions corresponding to the robot end effector and task-relevant objects. Random sampling is then performed within these regions. In real-world experiments, we leverage Grounded SAM 2 (Ravi et al., 2024; Ren et al., 2024) for segmentation. Prompted with "robot" and "white mouse", it can segment out the task-relevant part in the image observations. To improve the robustness of the resulting flow data, we apply this sampling strategy multiple times per image during both high-level and low-level training phases.

**Exploration Noise.**  During the online interaction process of our method, we add Gaussian noise with a standard deviation of 0.1 to each action dimension to encourage exploration. For the gripper dimension, which requires maintaining a consistent action value for multiple frames to close or open the gripper, we implement a custom exploration mechanism. This mechanism ensures that once the gripper transitions to a closed or open state, it remains in that state for a few frames, providing the necessary temporal consistency.

**Action chunking.**  Owing to the inherent challenges posed by temporally correlated confounders, a single-step policy is likely to encounter significant difficulties in achieving stable training. To address this issue and ensure stable policy training, we have adopted the techniques of action chunking and temporal ensemble, both of which have been widely utilized in prior research (Zhao et al., 2023; Chi et al., 2023; Zhao et al., 2024). Specifically, our flow-guided policy can be formally represented as $\pi_\theta(a_{t:t+k}|o_t, \mathbf{F}_{\text{flow}}(o_t, \mathbf{p}_t))$, as opposed to the conventional single-step policy formulation $\pi_\theta(a_t|o_t, \mathbf{F}_{\text{flow}}(o_t, \mathbf{p}_t))$. This implies that at every $k$ steps, the agent generates the subsequent $k$ actions based on the current observation and flow prediction. Furthermore, our temporal ensemble mechanism employs a weighted average over these predictions, utilizing an exponential weighting scheme defined as $w_i = \exp(-m \times i)$. In our implementation, the chunk size is consistently set to 5.

**Hyperparameters.**  In our flow prediction model, the training epoch is 1000, and the batch size is 64. The length and number of output flow are 16 and 32, respectively. The patch size of flow encoding is 4. The frame stack is set to be 1. During training, we randomly mask the image with a ratio of 0.5 for data augmentation, together with other data augmentations, including ColorJitter and random shift on flow to enhance robustness.

For the flow-conditioned policy training, the length and number of input flows are 8 and 32, respectively. The frame stack of the policy is set to be 2. The policy is pretrained with action-labeled data for 10000 iterations. The steps to interact with the environment and the steps to update the model are the same 80000 steps. The batch size for pretraining and online interaction is 64. During training, ColorJitter and random shift on flow are utilized to enhance robustness.

**Runtime and Computation Cost** We report the wall-clock time and computational resources required for our experiments to demonstrate the feasibility of HinFlow. All experiments were conducted on a single NVIDIA GeForce RTX 3090 GPU (24GB VRAM). For a single experimental seed, the pre-training stage takes approximately 30 minutes. The online imitation stage, which involves 80,000 environment interaction steps, takes approximately 11 hours.

## A.2 BASELINE DETAILS

**ATM (Wen et al., 2023):** We implement two baselines, ATM (grid) and ATM (seg), based on the ATM framework. For a fair comparison, both baselines utilize the same flow prediction model as our method for the high-level planner. ATM (grid) adheres to the original ATM architecture and hyperparameters. It employs a grid point sampling strategy, utilizing a fixed set of 32 points arranged on a grid. This baseline uses a frame stack of 10 and an input flow length of 16, which contrasts with our method's use of 2 and 8, respectively. During training, the image is first fed into the flow prediction model to obtain a predicted flow, which serves as input for the low-level policy. ATM (seg) incorporates our task-centric point sampling strategy. Like ATM (grid), it uses a frame stack of 10 and an input flow length of 16. However, during training, it directly uses ground-truth flow as input, aligning with our method.

**BC:** BC shares the same model architecture as ATM except that the predicted flow is zero-out in both training and evaluation. It uses a frame stack of 10 to optimize only a trivial image-action mapping.

**Online VPT:** The Online VPT contains two models: the inverse dynamics model and BC policy. The implementation of BC policy is detailed above. The implementation of IDM utilizes the spatial encoding and temporal encoding of the ATM architecture. The model first encodes each frame into a 64-dimensional spatial token. It is then passed into a temporal transformer. The original ATM adds a causal mask to the temporal transform. We can adapt it to an IDM by removing the causal mask. We choose a frame stack of 2 for IDM and BC policy. We also incorporate an action chunk of 5 for BC policy. The Online VPT updates IDM and BC every 10000 environment steps.

## B EXPERIMENT TASKS

**Task Details.** In this paper, we experiment with 4 tasks based on LIBERO (Liu et al., 2023) and 3 tasks based on ManiSkill3 (Tao et al., 2025). For each task, we set up a third-person camera and a wrist camera. The image resolution is $128 \times 128$. HinFlow samples points both on task-relevant objects and the robot gripper in the third-person camera and sample grid points in the wrist camera. Figure 9 and Figure 10 illustrate the third-party camera view for each task, and Figure 11 shows an example of the wrist camera in LIBERO and ManiSkill environments.

The proprioceptive states of LIBERO tasks are joint state and gripper state, and the proprioceptive state of ManiSkill tasks is end effector pose.

To mitigate the difficulty of predicting flow guidance caused by complex background variations during rotation, we restricted unnecessary rotational degrees of freedom based on the task requirements. For instance, only z-axis rotation was retained for the *Place Book* task, whereas all rotational degrees of freedom were disabled for all other tasks.

The per-task configurations for LIBERO tasks are as follows:

- *Place Butter:* The robot must pick up a stick of butter from the table and release it inside a basket. We sample 16 points on the butter and 16 points on the robot gripper. The horizon is 250.
- *Place Book:* The robot is required to grasp a book and lay it flat in the left compartment of a desk caddy. We sample 16 points on the book and 16 points on the robot gripper. The horizon is 200.
- *Hide Chocolate:* The robot is required to grasp a chocolate-pudding box, put it into the top drawer of a cabinet, and close the drawer. We sample 8 points on the chocolate-pudding, 8 points on the drawer and 16 points on the robot gripper. The horizon is 300.

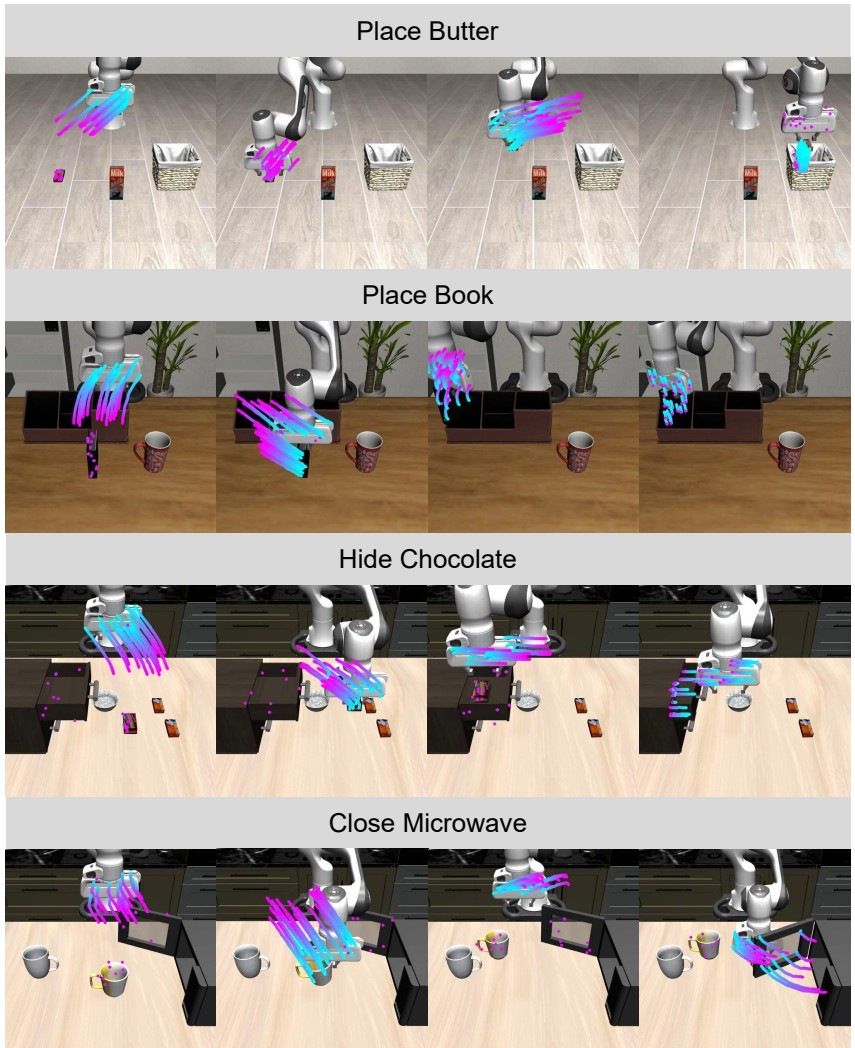

Figure 9: Visualizations of the 4 LIBERO tasks.

- ***Close Microwave:*** The robot must displace the yellow mug and then close the microwave door. We sample 8 points on the mug, 8 points on the microwave door and 16 points on the robot gripper. The horizon is 300.

The corresponding configurations for ManiSkill tasks are as follows:

- ***Place Sphere:*** The robot must pick up a sphere from the table and release it inside a bin. We sample 16 points on the robot gripper, 10 points on the sphere, and 6 points on the bin. The horizon is 150.
- ***Pull Cube Tool:*** The robot is required to use an L-shape stick to pull a cube far away. We sample 16 points on the robot gripper, 8 points on the stick and 8 points on the cube. The horizon is 250.
- ***Poke Cube:*** The robot grasps a stick to push a cube to a goal region. We sample 10 points on the robot gripper, 8 points on the stick, 8 points on the cube, and 6 points on the goal region. The horizon is 200.

**Data Collection.** We collect a large, diverse video dataset specifically for training a high-level planner. For each task, we gather an average of 300 demonstrations from existing datasets, scripted

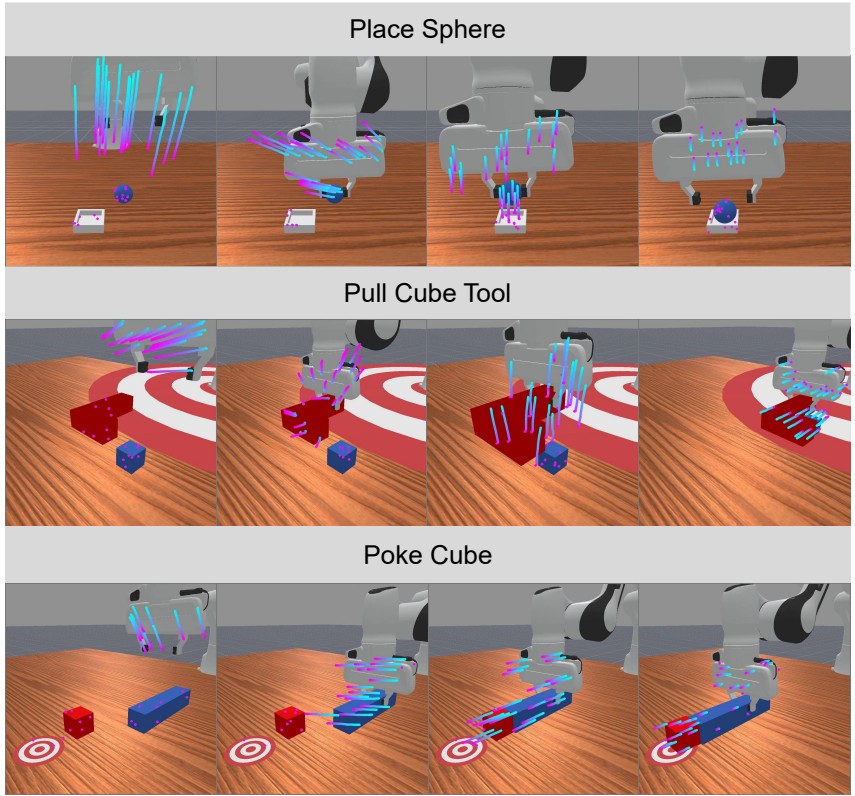

Figure 10: Visualizations of the 3 ManiSkill tasks.

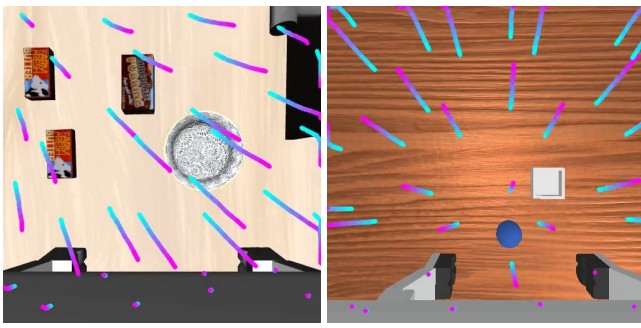

Figure 11: Example of wrist camera views in LIBERO and ManiSkill environments. Left: *Hide Chocolate* in LIBERO. Right: *Place Sphere* in ManiSkill.

policy, and SpaceMouse teleoperation. For some tasks in LIBERO, we also include relevant tasks in the same scene with a language token. We also collect the dataset beginning with diverse initial states to provide more reliable guidance to low-level policy.

## C  ADDITIONAL EXPERIMENT DETAILS

### C.1  CROSS-EMBODIMENT TRANSFER DETAILS

We perform cross-embodiment transfer experiments from Franka to Kinova in LIBERO and from Franka to xArm in ManiSkill. We collect 5 demonstrations for each task. The detail success rate curve is shown in Figure 12. We can observe that the online imitation may harm the overall performance if the high-level planner doesn't provide good flow guidance.

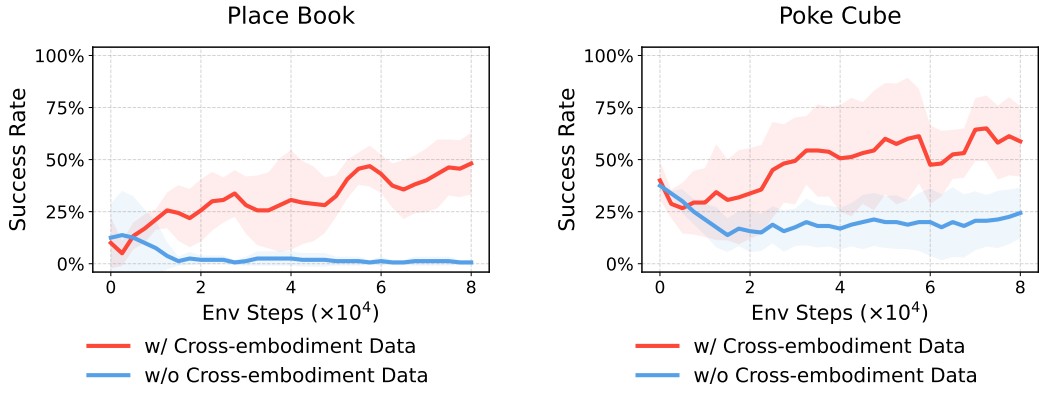

Figure 12: Comparison of HinFlow performance when high-level is trained with/without cross-embodiment data. Left: LIBERO Cross Embodiment. Right: ManiSkill Cross Embodiment.

## C.2 POLICY GENERALIZATION DETAILS

We implement two additional variants of the *Place Butter* task for policy generalization experiments. *Extra Distractors* is the standard scene augmented with many extra objects placed as visual distractors, and *Unseen Object* replaces the butter with an unseen chocolate-pudding box with a different color and size. In addition to the video dataset from the original task, we collect 500 video demonstrations for each generalization task using a script policy. Figure 13 visualizes these three experiments.

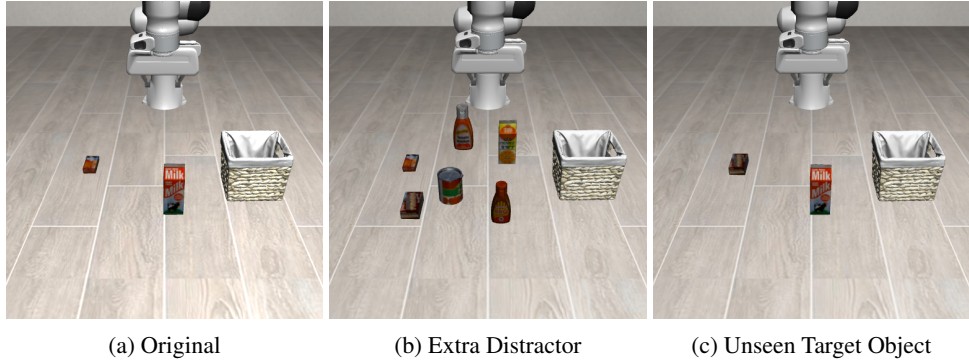

(a) Original      (b) Extra Distractor      (c) Unseen Target Object

Figure 13: Visualizations of policy generalization tasks.

## C.3 TASK UNSEEN TO BOTH PLANNER AND POLICY

We further conduct experiments where the task is unseen to both the high-level planner and the low-level policy. We modified the *Place Butter* task by replacing the butter's texture with 10 diverse variations, as illustrated in Figure 14. Both the high-level planner and the low-level policy are trained exclusively on these variations, using a dataset that contains 100 action-free videos and a single action-labeled demonstration per texture variant. The policy is then evaluated on the original *Place Butter* task. As shown in Table 3, HinFlow achieves a success rate of 85%, significantly outperforming the BC baseline. This demonstrates that our high-level planner generalizes well across visual domains, providing robust flow guidance for unseen tasks.

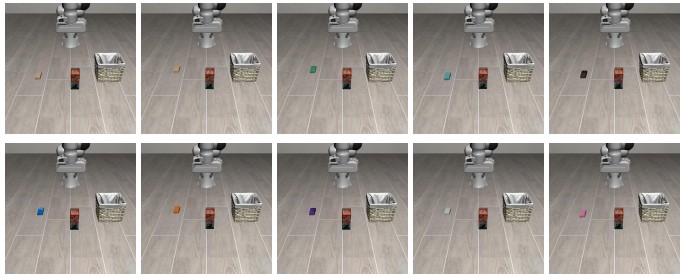

| Method | Success Rate(%) |
|--------|-----------------|
| BC | 36.0 |
| Ours | 85.0 |

Table 3: **Results on the unseen task.** Success rates evaluated on the original *Place Butter* task. Both methods are trained exclusively on the texture variations.

Figure 14: **Visualizations of the texture variants of *Place Butter* task.** We train the models on 10 diverse texture variants and evaluate them zero-shot on the original task.

## C.4 COMPARISON AGAINST ADDITIONAL BASELINES

### C.4.1 UNIPI

UniPi (Du et al., 2023) is a hierarchical method capable of effectively learning from action-free video data. It trains a video prediction model to serve as a high-level planner and learns an inverse dynamics model to infer actions from the videos. Since UniPi does not provide an open-source codebase, we follow the implementations in prior work (Ko et al., 2023; Wen et al., 2023) to reproduce this baseline.

The high-level planner is a diffusion-based generative model that predicts future frames (for both external and wrist cameras) as subgoals. The inverse dynamics model is a goal-conditioned policy $\pi(a_t, \text{done}_t | o_t, o_g)$, which predicts the action and termination signal given the current observation $o_t$ and the subgoal observation $o_g$. The "done" flag indicates whether the current subgoal has been achieved. During inference, the high-level planner generates seven future subgoals based on the initial observation and language description. At each step, the robot executes the action predicted by the inverse dynamics model and switches to the next subgoal if indicated by the done flag.

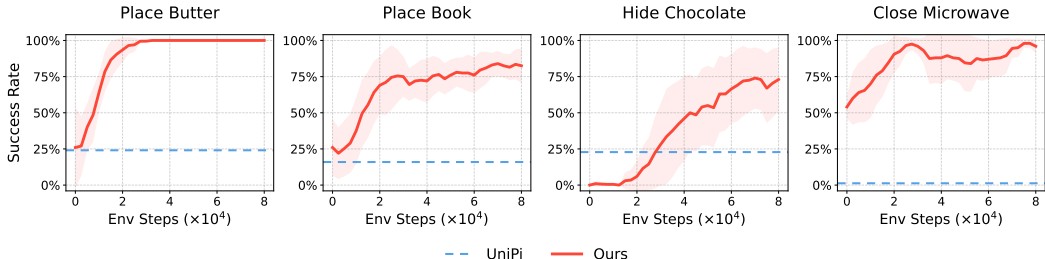

Figure 15: Comparison of HinFlow against UniPi baseline on the LIBERO tasks.

We conduct experiments on the LIBERO benchmark. For each task, we train an individual inverse dynamics model on a single action-labeled demonstration, while the high-level planner is a language-conditioned model trained on a large collection of unlabeled videos across all tasks. As shown in Figure 15, despite leveraging video data, UniPi fails to achieve a success rate exceeding 25% across all tasks. This is attributed to the scarcity of action-labeled demonstrations, which hampers the inverse dynamics model's ability to generate actions that accurately follow the high-level planner. In contrast, HinFlow empowers the robot to efficiently self-improve through online interaction, thereby enabling the grounding of high-level planning into a robust policy.

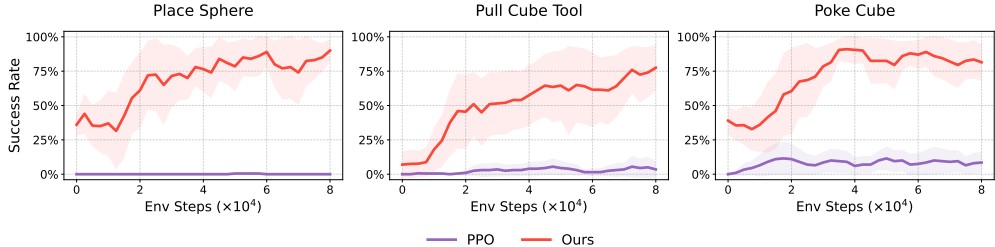

Figure 16: Comparison of HinFlow against PPO baseline on the ManiSkill tasks.

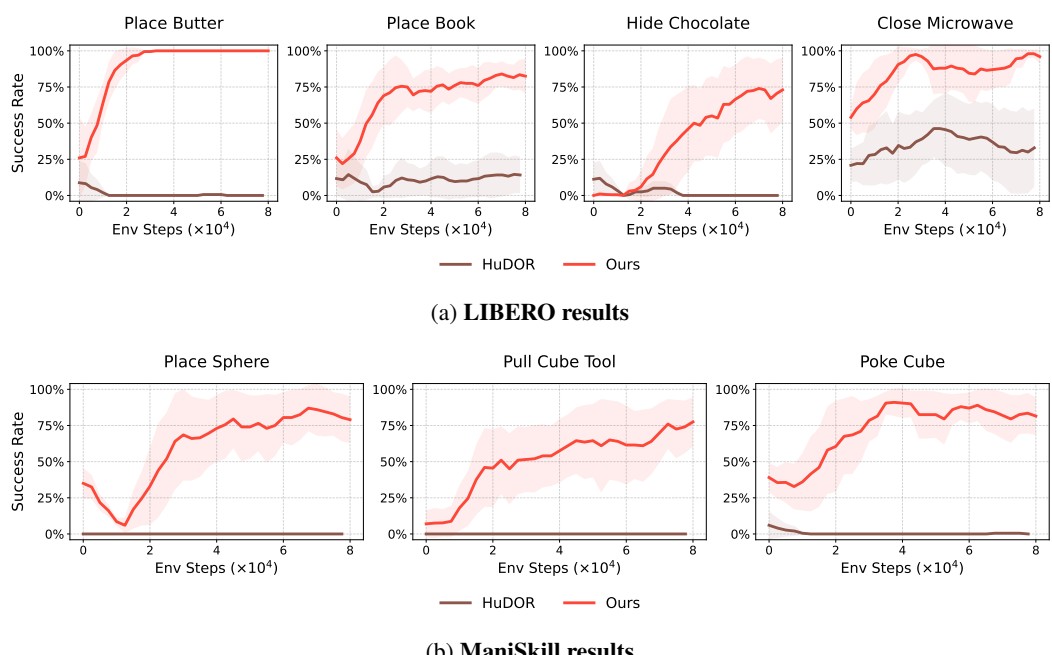

(a) **LIBERO results**

(b) **ManiSkill results**

Figure 17: Comparison of HinFlow against HuDOR.

### C.4.2 PPO

We include Proximal Policy Optimization (PPO) (Schulman et al., 2017) as a standard reinforcement learning baseline on the ManiSkill tasks. The experiments use the official RGB-based PPO implementation in ManiSkill with the provided dense reward function. As shown in Figure 16, PPO fails to solve these tasks in 80K environment steps: the success rate curve is below 10% on all three tasks. In contrast, HinFlow significantly improves the success rate within the same online interaction budget.

### C.4.3 HUDOR

In this section, we compare our method against HuDOR (Guzey et al., 2025), which uses object-oriented trajectory rewards to perform online residual reinforcement learning. HuDOR was originally developed for dexterous hand manipulation, so we made several modifications to adapt it to our setting. (1) While HuDOR uses human-hand-retargeted trajectories as the base policy, we replace that with the same pretrained policy used by our method. (2) To learn a residual policy via reinforcement learning, HuDOR relies on object-centric trajectories extracted from human videos for reward shaping. As an alternative, at each simulation reset, we generate an action-free success video from the current initial state using a scripted policy and provide it to HuDOR for reward computation. This is a form of privileged information that HinFlow does not receive. (3) We follow HuDOR's implementation for the residual policy architecture and RL algorithm. The residual policy's input is

a vector concatenating the base policy action, object-centric translational and rotational features, the robot's proprioceptive state, and the base policy's latent visual feature (which encodes observations from two cameras). The residual scale is 1.0 for the gripper action dimension and 0.05 for all other action dimensions. (4) Since the wrist-mounted camera cannot consistently observe the object, in both reward shaping and residual policy inputs, we only use the object-centric translational and rotational features from the third-person camera. (5) In addition, on top of the RL loss, we add an auxiliary behavior cloning loss computed from the action-labeled demonstrations, which we find improves the performance of the residual policy.

The results are shown in Figure 17. Our method significantly outperforms HuDOR across all tasks. HuDOR yields modest improvements over the base policy on *Place Book* and *Close Microwave*, but attains nearly zero success on the other five tasks. We suspect this is because HuDOR cannot effectively leverage the guidance in wrist-camera videos to reward residual policy learning. In contrast, HinFlow is more flexible and can be readily applied across different camera settings.

### C.5 MULTI-TASK FLOW PLANNER

To showcase the efficacy of HinFlow in multi-task settings, we trained a language-conditioned multi-task planner using the combined video dataset encompassing all tasks. In contrast to the task-specific training presented in our main results, this approach leverages the aggregated video data from all of our LIBERO tasks, where each task is associated with a unique language goal.

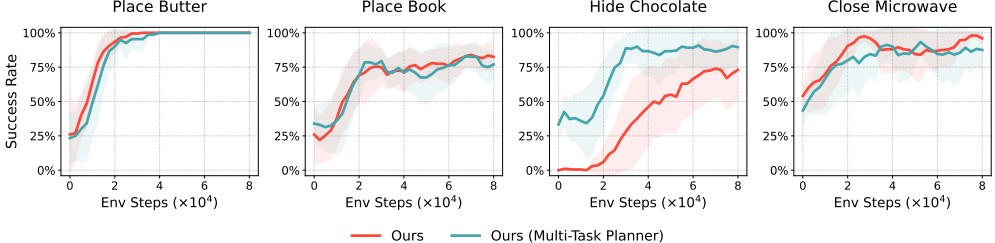

Figure 18: Results of multi-task planner.

As illustrated in Figure 18, this single planner achieves comparable performance on the tasks *Place Butter*, *Place Book*, and *Close Microwave*. Furthermore, it demonstrates a superior success rate on the *Hide Chocolate* task compared to the original, task-specific planner. These findings confirm that our methodology is capable of solving multiple tasks with a single, unified planner, leading to performance improvements.

### C.6 PERTURBATION ANALYSIS ON FLOW PLAN

To quantify HinFlow's sensitivity of our method to high-level prediction errors, we corrupt the planner-generated point flows with Gaussian noise during online data collection and evaluation. For each point's predicted trajectory $p_{t:t+H}$, we sample $(\Delta x, \Delta y) \sim \mathcal{N}(0, \sigma)$ and displace future point by

$$\Delta p_{t+i} = \frac{i}{H}(\Delta x, \Delta y), \quad i = 0, \ldots, H,$$

So the initial point remains unchanged while the entire trajectory receives a coherent directional bias. We conduct experiments under $\sigma \in \{8, 16, 24\}$ pixels. Note that the range of the point coordinate is $[0, 128]$. The example flows are visualized in Figure 19 and the results are shown in Figure 20.

When $\sigma = 8$ pixels, although the noise has already created a clear visual disturbance, performance on *Place Book*, *Hide Chocolate*, and *Place Sphere* shows almost no change, while performance on *Poke Cube* experiences a moderate decrease. This validates HinFlow's robustness against moderate planner errors. The success rate curve climbs slowly when $\sigma = 16$, and shows a mild decline when the noise is extremely large ($\sigma = 24$). The dominant failure mode arises at critical stages such as pick-up or release. In these states, inaccurate guidance causes the gripper to oscillate near key locations, precluding precise operation and leading to task failure.

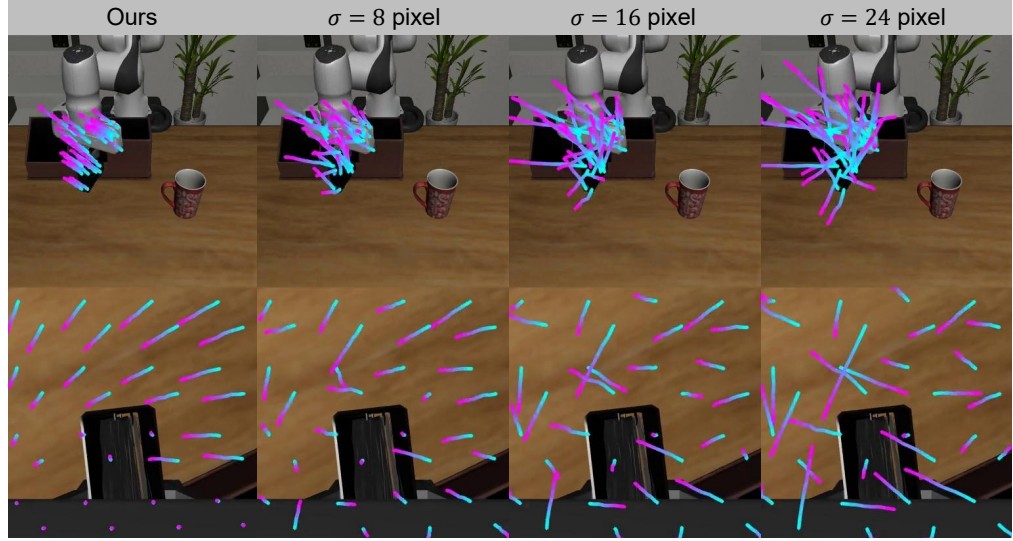

Figure 19: Example flows with different scales of Gaussian noise.

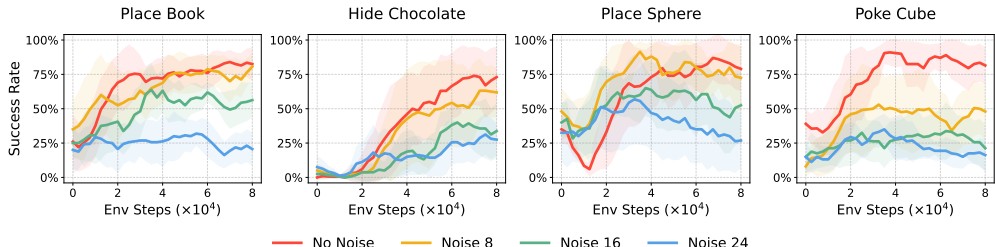

Figure 20: Results of perturbing flow plans with Gaussian noise.

### C.7 DISCUSSION OF PLACE SPHERE TASK

In our policy training process, we initialize the low-level policy using the available action-labeled expert demonstrations. Since the offline demonstrations have the same data structure, we can add these demonstrations into the replay buffer $\mathcal{D}_r$ to benefit the online training process. We evaluate this on *Place Sphere* Task. As shown in Figure 21, the success rate curve becomes smoother in the first 20K environment steps.

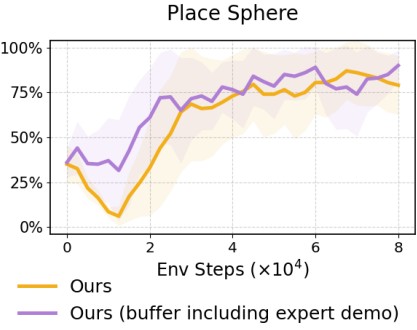

Figure 21: Results of *Place Sphere*, with expert demonstrations included in online replay buffer.

## C.8 Long Horizon Task

We construct a three-stage task in LIBERO in which the robot must sequentially pick up three distinct cuboid objects and place each of them into the basket (Figure 22a). The offline dataset consists of only two expert demonstrations produced by a scripted policy. Behavior cloning on this dataset yields a success rate of only 2% and places 0.7/3 objects on average. Initializing from the same pretraining dataset, HinFlow steadily improves performance, correctly placing 2.81 / 3 objects ultimately and confirming that flow-based hindsight imitation scales beyond two-stage horizons.

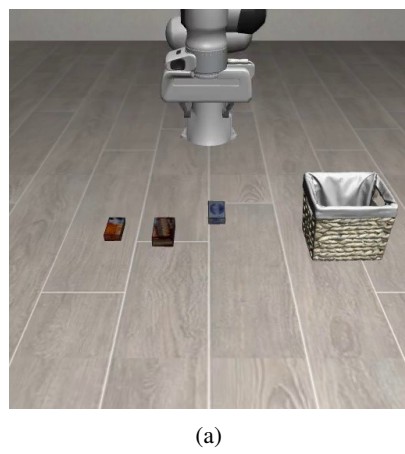
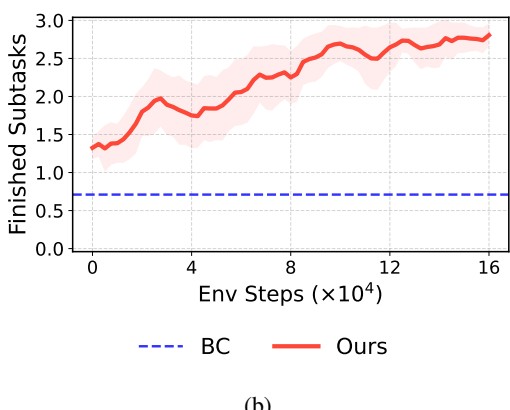

(a)                                              (b)

Figure 22: A three-stage long horizon task, where the robot must pick up three distinct objects and place each of them into the basket. Left: Visualization of the task configuration. Right: Experiment results.

## D LLM Usage

We used LLM to aid writing quality (grammar, phrasing, and organization). All LLM-generated text was reviewed, edited, and approved by the authors. The authors remain fully responsible for the paper's ideas, experiments, analyses, and conclusions.

