# OpenReview forum: "Translating Flow to Policy via Hindsight Online Imitation"
_ICLR.cc/2026/Conference — ICLR 2026 Poster_

### Official Review · Reviewer_kiqT · 2025-10-25

**Soundness:** 2
**Presentation:** 2
**Contribution:** 2
**Rating:** 4
**Confidence:** 3

**Summary:**

This paper presents HinFlow. First, it utilizes unlabeled robotic videos and off-the-shelf point trackers to train a high-level planner. Then, for the online interaction, by leveraging the thought of hindsight and regarding the generated flow trajectory as a replaceable goal, the low-level policy can improve itself without expert demonstrations. The author validates the effectiveness across 7 tasks and 2 benchmarks.

**Strengths:**

- Taking the high-level generated flow trajectory as a subgoal for low-level control is not novel, but regarding this as an replaceable goal in hindsight is interesting, and the experiments prove its effectiveness.
- This method is promising for scalable training because the high-level flow planner needs only action-free video, and the author validates that cross-embodiment data can help improve low-level policy learning.
- In section 5, HinFlow shows a very fast speed in convergence, which is promising to be applied in the real world for a fast policy improvement.

**Weaknesses:**

- The experiment is inadequate, with only 2 benchmarks and 7 tasks. Hope to see more results in the real world.
- Such kind of hierarchical methods cannot be used for large-data pretrained VLA models, such as $\pi_0$, Gr00t, RDT.
- The flow planner is task-specific and unable to deal with multi-task settings.
- The contribution is incremental.

**Questions:**

- Could HinFlow work as well in the real world as in simulators?
- Could HinFlow solve multiple tasks with only one flow planner model?
- All the experiments are at most 2 stages. What about using HinFlow in tasks with a longer horizon? Since a flow trajectory is always generated as an immediate subgoal, it may be suitable for complex tasks.
- Since there is no visualization of flow trajectories in wrist view, what's the role of the wrist view?

---

> ### Author Response · Authors · 2025-11-28
> **Response to Reviewer kiqT**
>
> Thank you for your detailed feedback! We conduct additional experiments to address your concerns and provide further clarifications to elaborate on the rationale behind our approach.
>
> > Q1: The experiment is inadequate, with only 2 benchmarks and 7 tasks. Hope to see more results in the real world.
>
> We thank the reviewer for the suggestion to strengthen our experimental evaluation. We have added a real-world experiment on a pick-and-place mouse task:
>
> | Method    | Success Rate |
> | --------- | ------------ |
> | BC        | 4/20         |
> | ATM (seg) | 8/20         |
> | **Ours**      | 8/20 → **19/20** |
>
> HinFlow significantly improved the success rate after collecting only 86 online interaction trajectories, demonstrating its robustness and efficiency in the physical world. Please refer to Section 5.3 for more details.
>
> > Q2: Such kind of hierarchical methods cannot be used for large-data pretrained VLA models, such as $\pi_0$, GR00T, RDT.
>
> We respectfully clarify that while these VLA models are typically end-to-end, our hierarchical framework is compatible with them and offers a potent strategy for fine-tuning. Specifically, a pretrained VLA can serve as the low-level policy in our framework by integrating flow guidance as an additional input modality. Furthermore, we envision that a large-scale Vision-Language Model (VLM) could act as the high-level planner to generate these flows. We argue that because such VLMs can be trained on a significantly broader scale of data than VLAs -- leveraging vast amounts of action-free videos alongside robotics data -- they hold the potential to demonstrate better robustness and generalization capabilities for novel tasks/scenarios. Consequently, our framework serves as a bridge that enables a potentially more capable VLM (which cannot generate executable actions) to guide the fine-tuning of the VLA. We view this work as a first step toward unlocking that promising future direction.
>
>
> > Q3: The flow planner is task-specific and unable to deal with multi-task settings. Could HinFlow solve multiple tasks with only one flow planner model?
>
> HinFlow is fully capable of solving multiple tasks using a single, unified flow planner model. To empirically validate this multi-task capability, we trained a single language-conditioned flow planner on the aggregated video dataset encompassing all our LIBERO tasks. The results, which are fully detailed in Appendix C.5, demonstrate the effectiveness of this approach. Specifically, the unified planner achieves comparable performance to the individual, task-specific planners on *Place Butter*, *Place Book*, and *Close Microwave*. Furthermore, it yields a superior success rate on the *Hide Chocolate* task, improving the average success rate from 71\% to 89\%.
>
> > Q4: All the experiments are at most 2 stages. What about using HinFlow in tasks with a longer horizon? Since a flow trajectory is always generated as an immediate subgoal, it may be suitable for complex tasks.
>
> We highlight that HinFlow is naturally suitable for longer-horizon tasks. To validate this point, we construct a three-stage task in LIBERO in which the robot must pick up three distinct cuboid objects and place each of them into the basket. We provide only 2 action-labeled demonstrations to HinFlow and the baseline. As shown in the following table, HinFlow ultimately learns to place 2.81/3 objects through online self-improvement. Please refer to Appendix C.8 for more details.
>
> | Method    | Finished Subtask |
> | --------- | --------------- |
> | BC        | 0.71             |
> | **Ours**  | **2.81**            |
>
> > Q5: Since there is no visualization of flow trajectories in wrist view, what's the role of the wrist view?
>
> Thank you for pointing out this detail. We would like to clarify that the wrist view does have corresponding flow trajectories. Since the target object may not consistently appear in the wrist view, we utilize a fixed set of grid points as query points for the high-level planner. We have now added wrist view flow visualizations to Figure 11 to make this explicit.

---

> > ### Comment · Reviewer_kiqT · 2025-11-28
> > **re**
> >
> > I appreciate the authors' detailed response to my comments. Based on these clarifications, I will raise my score to Borderline Accept -> 6.

---

> > > ### Author Response · Authors · 2025-11-28
> > >
> > > We sincerely appreciate your positive feedback and your decision to raise the score! If you have any further questions, please let us know. We would be happy to discuss them with you.

---

### Official Review · Reviewer_ReDe · 2025-10-27

**Soundness:** 3
**Presentation:** 4
**Contribution:** 3
**Rating:** 8
**Confidence:** 4

**Summary:**

This work proposes to use online interaction to improve the low-level policy that outputs action by taking in plans produced from high-level video-based planners. This is an important question since data with actions are much more expensive to obtain compared to video-only data. The experimental results shows promising performance of the proposed method in simulation environments (LIBERO, ManiSkill).

**Strengths:**

- The paper tackles a timely and important challenge in hierarchical robot learning: grounding high-level, video-derived plans into reliable low-level control.

- The proposed hindsight flow-conditioning idea is simple, intuitive, and well-motivated, yet leads to strong performance gains with good sample efficiency, achieving large improvements within 80K online interaction steps.

- The experiments span a diverse set of seven manipulation tasks across two widely used benchmarks (LIBERO and ManiSkill), highlighting generality.

- Cross-embodiment results are shows successful cross-robot skill transfer.

- The writing is clear and the paper is well-structured, making contributions easy to follow.

- Reproducibility is strong with detailed implementation descriptions and consistent reporting over multiple seeds.

**Weaknesses:**

While the reviewer is positive about the contributions and overall quality of this work, some clarifications and additions would further improve its clarity and impact:

- **Baseline comparison with non-hierarchical online learning.**
The current baselines emphasize either offline imitation (BC) or hierarchical approaches. Including a pure online RL baseline, such as PPO — even if it is expected to underperform within the 80K interaction budget — would help isolate the benefit of the flow-based hierarchical decomposition and better contextualize the results in Figure 6.

- **Clarification on environment reset assumptions.**
In real-world deployment, environment resets often incur human effort. The paper does not explicitly state the reset policy, though the figures suggest resets likely occur at the maximum task horizon. Given that noticeable performance gains emerge after ~20K environment steps (roughly 40–200 episodes, depending on the task), briefly discussing how this assumption affects practicality would be helpful.

- **Lack of real-robot experiments.**
Although the method shows strong robustness in simulation, real hardware introduces challenges — such as sensing latency, imperfect tracking, or actuator delay — that could impact the stability of the hindsight-labeling pipeline. A short discussion of this gap and potential mitigation strategies would strengthen the claims around deployability.

- **Runtime and compute cost reporting.**
The reproducibility section is detailed, but there is limited visibility into the wall-clock time or compute required for pretraining and online refinement. Even approximate training-time statistics (e.g., hours per stage on a given hardware setup) would make it easier for practitioners to gauge feasibility.

**Questions:**

- Could the authors comment on failure modes observed during online adaptation? For example, what happens when the high-level planner proposes inaccurate flow guidance?

- How sensitive is the approach to flow tracking noise during hindsight relabeling? Do the authors apply any filtering strategies to mitigate drift?

- One conceptual thought I had while reading: the hindsight relabeling mechanism seems naturally suited to situations where the robot remains in a continuous rollout without explicit resets. Even if the final state is not the intended goal, it could still be relabeled as a useful subgoal achieved along the way. That said, in these more open-ended scenarios, the robot may occasionally end up in unfamiliar states where the high-level planner struggles to provide sensible flow predictions. I’m curious how the authors view this possibility — do they see opportunities for the system to recognize and adapt to such out-of-distribution situations (e.g., self-recovery behaviors or gradual planner refinement)?

---

> ### Author Response · Authors · 2025-11-28
> **Response to Reviewer ReDe (Part I)**
>
> Thank you for your thorough and insightful comments! We are happy to hear that the reviewer appreciates the contributions of our work. We provide clarification to your concerns and questions as below:
>
> > Q1: Baseline comparison with non-hierarchical online learning. The current baselines emphasize either offline imitation (BC) or hierarchical approaches. Including a pure online RL baseline, such as PPO — even if it is expected to underperform within the 80K interaction budget — would help isolate the benefit of the flow-based hierarchical decomposition and better contextualize the results in Figure 6.
>
> Thanks for your suggestion. We include PPO as a standard reinforcement learning baseline on the ManiSkill tasks. The results are detailed in Appendix C.4.2. As expected, PPO yields success rates below 10\% for 80k environment steps on all three tasks, even with dense environmental rewards. This underscores the difficulty of pure online RL within a limited interaction budget and offers additional context for our results.
>
> > Q2: Lack of real-robot experiments. Although the method shows strong robustness in simulation, real hardware introduces challenges — such as sensing latency, imperfect tracking, or actuator delay — that could impact the stability of the hindsight-labeling pipeline. A short discussion of this gap and potential mitigation strategies would strengthen the claims around deployability.
>
> To address your concern, we have added a real-world experiment on a pick-and-place mouse task:
>
> | Method    | Success Rate |
> | --------- | ------------ |
> | BC        | 4/20         |
> | ATM (seg) | 8/20         |
> | **Ours**      | 8/20 → **19/20** |
>
> HinFlow significantly improved the success rate after collecting only 86 online interaction trajectories, demonstrating its reliability in the physical world. Regarding your concern about the system delay, we have optimized our pipeline to run at 10 FPS stably.
> We also constrain the robot’s movement speed to ensure the background changes remain smooth in the wrist view, preventing CoTracker3 from generating large tracking errors. Please refer to Section 5.3 for more details.
>
> > Q3: Clarification on environment reset assumptions. In real-world deployment, environment resets often incur human effort. The paper does not explicitly state the reset policy, though the figures suggest resets likely occur at the maximum task horizon. Given that noticeable performance gains emerge after ~20K environment steps (roughly 40–200 episodes, depending on the task), briefly discussing how this assumption affects practicality would be helpful.
>
> In both simulation and real-world experiments, we reset the environment upon task success or reaching the maximum horizon. While in real-world experiments, human effort is required to reposition the object. This setup is consistent with standard assumptions in recent real-world reinforcement learning literature [1,2]. We agree that reducing human intervention is crucial, and we view integrating our method with research on reset-free policy learning [3] or automated reset techniques [4] as an important direction to further enhance practicality.
>
> > Q4: Runtime and compute cost reporting. The reproducibility section is detailed, but there is limited visibility into the wall-clock time or compute required for pretraining and online refinement. Even approximate training-time statistics (e.g., hours per stage on a given hardware setup) would make it easier for practitioners to gauge feasibility.
>
> We thank the reviewer for this suggestion. We have included a detailed computational cost report in Appendix A.1. For a single experimental seed, the pretrain stage takes approximately 30 minutes and the online imitation stage (80,000 interaction steps) takes approximately 11 hours on a single NVIDIA RTX 3090 GPU.
>
>
> [1] Wagenmaker, Andrew, et al. "Steering Your Diffusion Policy with Latent Space Reinforcement Learning." Conference on Robot Learning. 2025.
>
> [2] Lei, Kun, et al. "RL-100: Performant Robotic Manipulation with Real-World Reinforcement Learning." arXiv preprint arXiv:2510.14830 (2025).
>
> [3] Gupta, Abhishek, et al. "Reset-free reinforcement learning via multi-task learning: Learning dexterous manipulation behaviors without human intervention." 2021 IEEE International Conference on Robotics and Automation (ICRA). IEEE, 2021.
>
> [4] Luo, Jianlan, et al. "Serl: A software suite for sample-efficient robotic reinforcement learning." 2024 IEEE International Conference on Robotics and Automation (ICRA). IEEE, 2024.

---

> ### Author Response · Authors · 2025-11-28
> **Response to Reviewer ReDe (Part II)**
>
> > Q5: Could the authors comment on failure modes observed during online adaptation? For example, what happens when the high-level planner proposes inaccurate flow guidance?
>
> The dominant failure mode happens in the key stages, such as pick-up or place, which require precise alignment and operation. In Appendix C.6, we corrupt the planner-generated point flows with Gaussian noise to simulate inaccurate flow guidance. We observe that in the critical stages, perturbed guidance causes the gripper to oscillate near key locations, precluding precise operation and derailing task execution.
>
> > Q6: How sensitive is the approach to flow tracking noise during hindsight relabeling? Do the authors apply any filtering strategies to mitigate drift?
>
> Our approach is robust to typical tracking noise during hindsight relabeling, relying on two strategies without explicit filtering: (1) Apply random shift augmentation to flow inputs during training the low-level policy. This simulates tracking errors, ensuring the policy remains robust even if hindsight labels contain noise. (2) Control tracking quality at the source. As discussed in Q2, in real-world experiments, we constrain the robot's movement speed to avoid large tracking errors.
>
> > Q7: One conceptual thought I had while reading: the hindsight relabeling mechanism seems naturally suited to situations where the robot remains in a continuous rollout without explicit resets. Even if the final state is not the intended goal, it could still be relabeled as a useful subgoal achieved along the way. That said, in these more open-ended scenarios, the robot may occasionally end up in unfamiliar states where the high-level planner struggles to provide sensible flow predictions. I’m curious how the authors view this possibility — do they see opportunities for the system to recognize and adapt to such out-of-distribution situations (e.g., self-recovery behaviors or gradual planner refinement)?
>
> We fully agree with the reviewer's insight. Hindsight relabeling makes our low-level policy learning naturally compatible with reset-free settings. However, to fully support continuous rollout settings, the high-level planner must possess error-recovery capabilities. We envision that a large-scale vision-language model (VLM) could serve as this high-level planner in the future. When trained on massive amounts of video data, such a VLM may naturally exhibit emergent error-recovery behaviors and produce sensible flow predictions even in unfamiliar states. To further enhance this capability, we can prompt the VLM with a video clip of the agent’s pre-OOD trajectory to generate flows that guide the robot back toward in-distribution states. We view this as a promising direction for future work.

---

### Official Review · Reviewer_EHLv · 2025-11-01

**Soundness:** 3
**Presentation:** 2
**Contribution:** 2
**Rating:** 4
**Confidence:** 3

**Summary:**

The paper presents HinFlow, a framework that grounds the point flow predicted from a high-level planner to executable actions using hindsight relabeling. Specifically, as data for flow-conditioned low-level policy training can be expensive to curate, the method deploys the low-level policy in the environment to collect exploratory rollouts, which can be annotated with flow labels using a video tracker. The relabeled data can then be used to update the policy. Evaluations conducted on two robotic manipulation benchmarks showcase the effectiveness of the method, even with environmental perturbations.  Furthermore, the framework enables cross-embodiment generalization, underscoring the robustness of point flow representations.

**Strengths:**

- The proposed method effectively mitigates the data scarcity problem of grounding point flow to environmental actions via hindsight relabeling.
- The extensive evaluations show that the framework can not only iteratively refine with relabeled data compared to the baselines, but also release the full potential of flow representation in task and embodiment generalization.
- Single-task policy can converge to high performance with a small number of environment interactions, highlighting the sample efficiency of the method.

**Weaknesses:**

- Both point flow representations and hindsight relabeling have been widely investigated by prior works for generalizability and resolving data scarcity issues, respectively. For video/flow-based methods, more baselines should be considered, like UniPi [1] and FlowDiffusion [2]. Also, it would be helpful to explain why in the Place Sphere task, the performance has a decreasing-and-increasing trend.
- The success of the proposed method heavily depends on the flow quality predicted by the high-level planner. When evaluating cross-embodiment and task generalization, no task is truly novel for high-level planners. It is also important to evaluate the robustness of the framework as a whole when handling tasks unseen to both the high-level planner and the low-level policy.
- High-level planners can be trained on video or action-free demonstrations, which ideally should benefit more from large-scale web videos from the real world for better generalizability. On the other hand, deploying an initially unstable policy for exploration and data collection can be very dangerous or costly on real robots, limiting the real-world applications of hindsight relabeling.

[1] Du et al. Learning Universal Policies via Text-Guided Video Generation. NeurIPS 2023.

[2] Ko et al. Learning to Act from Actionless Videos through Dense Correspondences. 2023.

**Questions:**

See above.

---

> ### Author Response · Authors · 2025-11-28
> **Response to Reviewer EHLv (Part I)**
>
> Thank you for your valuable comments! We conduct supplementary experiments and provide detailed clarifications to address your concerns.
>
> > Q1: For video/flow-based methods, more baselines should be considered, like UniPi [1] and FlowDiffusion [2].
>
> We thank the reviewer for suggesting these relevant baselines. Regarding UniPi: As shown in the following table, UniPi's performance is low despite leveraging video data. We attribute this to the scarcity of action-labeled demonstrations, which severely hampers the inverse dynamics model's ability to generate actions that accurately follow the high-level planner. Please refer to Appendix C.4.1 for more details.
>
> | Task             | UniPi | **Ours**            |
> |------------------|-------|----------------------|
> | Place Butter     | 16%   | 26% → **100%**       |
> | Place Book       | 24%   | 26% → **82.5%**      |
> | Hide Chocolate   | 23%   | 0% → **73%**         |
> | Close Microwave  | 1%    | 54% → **96%**        |
>
> Regarding FlowDiffusion (AVDC): We did not include this baseline as it imposes additional input requirements (e.g., depth map of the initial frame) that are inconsistent with our problem setup. Instead, we compared against state-of-the-art 2D flow-based methods (ATM and its variants) in our main experiments.
>
> Most importantly, we emphasize that HinFlow's key advantage over these offline baselines lies in its ability to efficiently self-improve through online interaction, thereby enabling the grounding of high-level planning into a robust policy.
>
> > Q2: It would be helpful to explain why in the Place Sphere task, the performance has a decreasing-and-increasing trend.
>
> The *Place Sphere* task requires precise manipulation: the gripper must center on the sphere and then align it with a bin of comparable size before releasing. At the start of online imitation, the exploration process will produce a high proportion of failed attempts. Consequently, the model updates on a buffer dominated by unsuccessful trajectories, causing a distribution mismatch with the pretraining dataset that temporarily degrades performance. To mitigate this instability, we incorporate pretraining data into the buffer. As reported in Appendix C.7, this simple change removes the decreasing trend and yields more stable performance.
>
> > Q3: When evaluating cross-embodiment and task generalization, no task is truly novel for high-level planners. It is also important to evaluate the robustness of the framework as a whole when handling tasks unseen to both the high-level planner and the low-level policy.
>
> First, we believe that scaling up the high-level planner on larger, more diverse datasets will fundamentally improve generalization, allowing the model to predict flows for truly novel tasks zero-shot. While our current model size and dataset are limited, we conducted a controlled experiment to preliminarily validate this generalization potential (see Appendix C.3). Specifically, we created 10 distinct texture variations of the Place Butter task. We trained both the high-level planner and the low-level policy exclusively on these 10 variations (using a dataset of 100 action-free videos and 1 labeled demonstration per variation). We then evaluated the policy on the original Place Butter task:
>
> | Method    | Success Rate(%) |
> | --------- | --------------- |
> | BC        | 36.0            |
> | **Ours**      | **85.0**            |
>
> HinFlow significantly outperforms the BC baseline. This demonstrates that our high-level planner can extract generalized flow representations to guide the policy through novel visual scenarios.

---

> > ### Author Response · Authors · 2025-11-28
> > **Response to Reviewer EHLv (Part II)**
> >
> > > Q4: Deploying an initially unstable policy for exploration and data collection can be very dangerous or costly on real robots, limiting the real-world applications of hindsight relabeling.
> >
> > To address your concern, we have conducted a real-robot experiment on a pick-and-place mouse task:
> >
> > | Method    | Success Rate |
> > | --------- | ------------ |
> > | BC        | 4/20         |
> > | ATM (seg) | 8/20         |
> > | **Ours**      | 8/20 → **19/20** |
> >
> > HinFlow significantly improved the success rate after collecting only 86 online interaction trajectories, validating that our method can be highly sample-efficient on real hardware. Regarding safety concerns, we emphasize that online exploration in physical environments is a standard setting in real-world reinforcement learning literature [1,2]. Following common practice, we reduced safety issues by: (1) constraining the workspace to avoid collisions with the tabletop and surrounding devices, (2) capping maximum speeds to prevent abrupt motions, and (3) adding pretraining demonstrations to the online replay buffer to stabilize early policy updates (as discussed in Q2). Please refer to Section 5.3 for more details.
> >
> > [1] Wagenmaker, Andrew, et al. "Steering Your Diffusion Policy with Latent Space Reinforcement Learning." Conference on Robot Learning. 2025.
> >
> > [2] Lei, Kun, et al. "RL-100: Performant Robotic Manipulation with Real-World Reinforcement Learning." arXiv preprint arXiv:2510.14830 (2025).

---

### Official Review · Reviewer_6MXz · 2025-11-01

**Soundness:** 2
**Presentation:** 3
**Contribution:** 2
**Rating:** 6
**Confidence:** 3

**Summary:**

The paper proposes a hierarchical robot-learning framework, HinFlow that uses a flow-based high-level planner. The key idea is hindsight relabeling on flows during rollouts and aggregating them back to the data buffer for policy training. The paper shows 84% SR in seven manipulation tasks from LIBERO and ManiSkill using 80K online interaction steps.

**Strengths:**

The key idea is hindsight relabeling of achieved flows for dense supervision during online practice is well used data augmentation technique that enable sample-efficient policy training.
The experiments highlight robustness and versatility for zero-shot generalization to novel objects/distractors when the planner covers those visuals.
The paper motivates flows as a compact, appearance-robust high-level representation and positions HinFlow as a simple bridge that grounds those plans into executable policies via online imitation.

**Weaknesses:**

The proposed approach of hindsight relabeling depends on the flow quality and point tracking/segmentation. Since the results are in sim where this feasible, it is unclear if the approach is robust to sim2real gaps and reliably work in real world.
The point tracking is challenging also due to 2D ambiguity for 3D motions, occlusions and high-speed motions.

Policy trains on achieved flows but infers under predicted flows and a potential distribution shift has not been analyzed.

**Questions:**

How sensitive is HinFlow to planner accuracy? Do you have thresholds where online imitation starts hurting? What are the dominant failure modes?
What’s the most straightforward path to 3D motion fields (e.g., point clouds, SE(3) flows)? How would hindsight relabeling extend to 3D?
The evaluation focuses on tabletop pick-n-place tasks, with language conditioning. How much does language help multi-tasking or transfer? Does the language input need to be changed with hindsight relabelled trajectories for policy training?

---

> ### Author Response · Authors · 2025-11-28
> **Response to Reviewer 6MXz (Part I)**
>
> Thank you for your thoughtful feedback! We conduct additional experiments to answer your questions and discuss possible future directions below.
>
> > Q1: Since the results are in sim where this feasible, it is unclear if the approach is robust to sim2real gaps and reliably work in real world. The point tracking is challenging also due to 2D ambiguity for 3D motions, occlusions and high-speed motions.
>
> We highlight that, rather than relying on sim2real transfer, HinFlow's high sample efficiency makes it suitable for online improvement on real robots. We have conducted a real-robot experiment on a pick-and-place mouse task:
>
> | Method    | Success Rate |
> | --------- | ------------ |
> | BC        | 4/20         |
> | ATM (seg) | 8/20         |
> | **Ours**      | 8/20 → **19/20** |
>
> HinFlow significantly improved the success rate after collecting only 86 online interaction trajectories, demonstrating its reliability in the physical world. Regarding your concerns on technical issues: (1) We use a third-view camera together with a wrist-mounted camera. This combination effectively mitigates issues arising from 2D ambiguity and occlusions. (2) We constrain the robot's movement speed to ensure the background changes remain smooth in the wrist view, preventing CoTracker3 from generating large tracking errors. Please refer to Section 5.3 for more details.
>
>
> > Q2: Policy trains on achieved flows but infers under predicted flows and a potential distribution shift has not been analyzed.
>
> We acknowledge the distribution shift between training on achieved flows and inferring with predicted flows. Training on achieved flows is necessary to ensure the low-level policy learns accurate conditional dynamics. To mitigate the impact of this inherent shift, we apply data augmentation to the achieved flows during policy training. Please also see Q3, where we discuss the impact of the planner's accuracy in detail.
>
>
> > Q3: How sensitive is HinFlow to planner accuracy? Do you have thresholds where online imitation starts hurting? What are the dominant failure modes?
>
> To quantify HinFlow’s sensitivity to high-level prediction errors, we corrupt the planner-generated point flows with Gaussian noise. We conduct experiments under $ \sigma = 8,16,24$ pixels, while the image resolution is $128\times 128$. As shown in Appendix C.6, the perturbation is visually obvious at $\sigma = 8$ pixels, yet the success rates for *Place Book*, *Hide Chocolate*, and *Place Sphere* remain almost unchanged; only *Poke Cube* shows a moderate decline. Learning process slows down at $\sigma=16$ pixels, and extremely large noises ($\sigma=24$ pixels) may cause a slight performance drop during training. The dominant failure mode arises at critical stages such as pick-up or release. In these states, inaccurate guidance causes the gripper to oscillate near key locations, precluding precise operation and leading to task failure.

---

> > ### Author Response · Authors · 2025-11-28
> > **Response to Reviewer 6MXz (Part II)**
> >
> > > Q4: What’s the most straightforward path to 3D motion fields (e.g., point clouds, SE(3) flows)? How would hindsight relabeling extend to 3D?
> >
> > We appreciate this insightful question. A straightforward path to 3D is to use the trajectories of a point set in 3D space as the high-level plan. This can be implemented by either (1) directly training the planner on RGB-D datasets [1] and using depth cameras for perception in inference, or (2) maintaining 2D image inputs while leveraging visual trackers capable of lifting 2D pixels into 3D (e.g., SpatialTracker [2]) to extract 3D flows. Subsequently, extending hindsight relabeling to this setting is quite intuitive: we can employ the tracker to compute the achieved 3D flows from the agent's online rollouts and simply treat them as goal labels to supervise the low-level policy training.
> >
> > Our experiments and related work [3,4,5] show that 2D point flows have already enabled effective use of video data and yielded substantial improvements in sample efficiency. As 3D vision models continue to scale up and improve in precision, we believe that extending the core ideas of HinFlow to 3D has strong potential to further improve the understanding and generalization of complex 3D manipulation.
> >
> > > Q5: The evaluation focuses on tabletop pick-n-place tasks, with language conditioning. How much does language help multi-tasking or transfer? Does the language input need to be changed with hindsight relabelled trajectories for policy training?
> >
> > For LIBERO scenes that share similar initial configurations but differ in task objectives, we augment the video demonstrations with a language description that unambiguously specifies the intended goal (e.g., “place the book in the left/right/front/back compartment”). This linguistic disambiguation enables the high-level planner to leverage a broader set of video data while remaining on the correct task target. For the low-level part, the policy receives only visual and proprioceptive inputs, without any linguistic input.
> >
> > [1] Yuan, Chengbo, et al. "General Flow as Foundation Affordance for Scalable Robot Learning." Conference on Robot Learning. PMLR, 2025.
> >
> > [2] Xiao, Yuxi, et al. "Spatialtracker: Tracking any 2d pixels in 3d space." Proceedings of the IEEE/CVF Conference on Computer Vision and Pattern Recognition. 2024.
> >
> > [3] Bharadhwaj, Homanga, et al. "Track2act: Predicting point tracks from internet videos enables generalizable robot manipulation." European Conference on Computer Vision. Cham: Springer Nature Switzerland, 2024.
> >
> > [4] Wen, Chuan, et al. "Any-point Trajectory Modeling for Policy Learning." Proceedings of Robotics: Science and Systems XX, 2024.
> >
> > [5] Xu, Mengda, et al. "Flow as the Cross-domain Manipulation Interface." Conference on Robot Learning. PMLR, 2025.

---

### Author Response · Authors · 2025-12-01
**To AC: Summary of Our Efforts During Rebuttal**

Dear Area Chair,

We sincerely appreciate the time and effort dedicated by all reviewers. We are highly encouraged by their recognition of our work's strengths, particularly in two key aspects:

- **Problem & Motivation**: Reviewers commended the high impact of our work, agreeing that it "tackles a timely and important challenge in hierarchical robot learning" *(ReDe)*. This position was compellingly summarized as "a simple bridge that grounds plans into executable policies." *(6MXz)* Complementing this, other reviewers acknowledged that our framework "effectively mitigates the data scarcity problem" *(EHLv)* and is "promising for scalable training" *(kiqT)*.

- **Methodology & Experimental Verification**: The proposed method was praised as "simple, intuitive, and well-motivated" *(ReDe)*, and the mechanism of treating flow trajectory as a replaceable goal in hindsight was considered "interesting" *(kiqT)*. Our effective design results in high sample efficiency, a key strength consistently recognized by *all four reviewers*. Furthermore, reviewers explicitly highlighted the method’s potential to "release the full potential of flow representation in task and embodiment generalization" *(EHLv)* and its demonstrated "robustness and versatility for zero-shot generalization to novel objects/distractors when the planner covers those visuals" *(6MXz)*.

During the rebuttal phase, we engaged in productive discussions with the reviewers and conducted substantial additional work to address their concerns. All revisions in the updated manuscript have been marked in blue. Key highlights of these discussions include:

- **Addressing real-world deployability concerns** *(6MXz Q1, EHLv Q4, ReDe Q2, kiqT Q1)*.
We conducted a new real-world experiment on a mouse pick-and-place task, which has been added to Section 5.3. HinFlow improves the success rate from 40\% to 95\% within only 86 online interaction trajectories (~1 hour of interleaved data collection and model updates), providing strong empirical evidence of its robustness and efficiency in the physical world.

- **Analyzing impacts of inaccurate flow plans** *(6MXz Q3, ReDe Q5)*.
In Appendix C.6, we introduced a new experiment where we corrupted the planner-generated point flows with random noise to simulate inaccurate guidance. We observed that HinFlow's performance remains robust when the noise is at a reasonable scale. We also discussed the failure modes under excessive noise levels.

- **Discussions on future work**.
Regarding the reviewers' insightful questions, we provided detailed discussions on future directions, specifically covering: (1) potential pathways for extending HinFlow to 3D flow plans *(6MXz Q4)*, (2) the feasibility of extending HinFlow to continuous rollout settings *(ReDe Q7)*, and (3) the possibility of fine-tuning VLAs based on the HinFlow framework *(kiqT Q2)*.

- **Evaluations on more challenging tasks**.
We added experiments covering two challenging scenarios: (1) a novel task unseen by both the planner and the policy, detailed in Appendix C.3 *(EHLv Q3)*, and (2) a three-stage long-horizon task, detailed in Appendix C.8 *(kiqT Q4)*. Our method successfully solves both challenges.

- **New implementation-level improvements**.
We found that (1) incorporating pretrained demonstrations into the replay buffer leads to more stable performance *(EHLv Q2)*, and (2) training a single, unified flow planner model using data from all tasks further enhances success rates *(kiqT Q3)*. The discussions are included in Appendix C.7 & C.5.

- **Comparison with more baselines**.
Following the reviewers' suggestions, we further compared our method with UniPi *(EHLv Q1)* and PPO *(ReDe Q1)*. As shown in Appendix C.4, HinFlow significantly outperforms these baselines.

- **Clarifications on other experimental details** *(6MXz Q2 & Q5, ReDe Q3 & Q4 & Q6, kiqT Q5)*.

We believe that our rebuttal materials have resolved all the questions raised by the reviewers. We are also delighted to note that Reviewer kiqT has already acknowledged our response and increased the score. Unfortunately, the other reviewers were unable to respond due to the system locking. We respectfully hope that you will take these circumstances into consideration, and we would be grateful if the significant improvements we achieved during the rebuttal could be factored into your final decision.

Finally, we sincerely thank you for your commitment to upholding the integrity and fairness of the ICLR review process. We deeply appreciate the considerable effort you have dedicated to handling our paper under these challenging circumstances.


Sincerely,

The Authors

---

### Meta-Review · Area_Chair_zzrr · 2026-01-09

**Summary:**

The paper presents a framework that grounding point flow predictions from a high-level planner to a low level policy using hindsight relabeling. As data for flow-conditioned low-level policy training can be expensive to curate, the method deploys the low-level policy in the environment to collect exploratory rollouts, which can be annotated in hindsight with flow labels using a video tracker. The relabeled data can then be used to update the policy. They evaluate this methodology in sim and in real for robotics problems .

Some primary reviewer concerns were:
1) Real-world robustness / sim-to-real feasibility is unproven
2) Heavy dependence on high-level planner + tracker quality (sensitivity not analyzed)
3) Novelty / contribution perceived as incremental
4) Baselines and evaluation breadth are considered insufficient by some
5) Clarifications missing about experimental setup
6) Positioning vs large pretrained VLA models

**Reviewer Concerns:**

The rebuttal addressed most of the concerns raised by reviewers. There are some remaining ideological concerns, but I think those are biases of the reviewers rather than fundamental commentary on the method. The paper as constructed addresses most significant reviewer concerns, they added a real robot experiment, analysis on noisy flow paths, more baselines and further experimental details and analysis. They also added some further stability and usability improvements to their method.

**Reviewer Scores:**

Reviewer 6MXz would likely have raised from 6 to 7 since questions were addressed and real world experiments added.
Reviewer EHLv would likely have raised from 4 to 6 since the additional baselines were added.
Reviewer ReDe already accepted and would likely keep the same.
Reviewer kiqT would maybe not raise or raise to a 5. But I think their review comments were a bit uninformative and too high level. I think the ideological take her should be discarded.

---

### Decision · Program_Chairs · 2026-01-26

Accept (Poster)